# High-resolution neural recordings improve the accuracy of speech decoding

Suseendrakumar Duraivel[1], Shervin Rahimpour[2,3], Chia-Han Chiang[1], Michael Trumpis[1], Charles Wang[1], Katrina Barth [1], Stephen C. Harward [2,4], Shivanand P. Lad[2], Allan H. Friedman[2], Derek G. Southwell[1,2,4,5], Saurabh R. Sinha[6], Jonathan Viventi [1,2,4,5] ✉ & Gregory B. Cogan [1,2,4,7,8,9] ✉

Patients suffering from debilitating neurodegenerative diseases often lose the ability to communicate, detrimentally affecting their quality of life. One solution to restore communication is to decode signals directly from the brain to enable neural speech prostheses. However, decoding has been limited by coarse neural recordings which inadequately capture the rich spatio-temporal structure of human brain signals. To resolve this limitation, we performed high-resolution, micro-electrocorticographic (μECoG) neural recordings during intra-operative speech production. We obtained neural signals with 57× higher spatial resolution and 48% higher signal-to-noise ratio compared to macro-ECoG and SEEG. This increased signal quality improved decoding by 35% compared to standard intracranial signals. Accurate decoding was dependent on the high-spatial resolution of the neural interface. Non-linear decoding models designed to utilize enhanced spatio-temporal neural information produced better results than linear techniques. We show that high-density μECoG can enable high-quality speech decoding for future neural speech prostheses.

Motor disorders such as amyotrophic lateral sclerosis (ALS) and clinical locked-in syndromes greatly reduce or eliminate patients' ability to verbally communicate and dramatically affect their quality of life. There are ~6000 new ALS diagnoses in the United States each year, resulting in near-complete loss of motor function but preserving cognitive functions[1,2]. Computer aided technologies have been reported to improve quality of life but are often limited by slow processing and inefficiencies[3–5]. Neural speech prostheses offer the potential for a faster and more reliable means to decode conversational speech directly from the brain. The development of high-resolution neural recordings would enable accurate decoding of spoken features which is paramount to a successful neural speech prosthesis.

Previous attempts to accurately decode speech have typically utilized invasive macro electrocorticography (macro ECoG, 10 mm inter-electrode spacing and 2.3 mm exposed diameter), and high-density ECoG (4 mm inter-electrode spacing and 1 mm exposed diameter), or stereo-electroencephalography (SEEG, 3.5 – 5 mm inter-electrode spacing), that target ventral sensorimotor cortex or speech motor cortex (SMC) during speech production[6–10]. These studies demonstrated that SMC encodes articulatory properties of speech motor sequences which can form the building blocks of successful speech decoding. These articulatory features are subsequently transformed into acoustic speech[11–13] or can be combined to form fundamental linguistic units such as phonemes[14–19], which can be aggregated

[1]Department of Biomedical Engineering, Duke University, Durham, NC, USA. [2]Department of Neurosurgery, Duke School of Medicine, Durham, NC, USA. [3]Department of Neurosurgery, Clinical Neuroscience Center, University of Utah, Salt Lake City, UT, USA. [4]Duke Comprehensive Epilepsy Center, Duke School of Medicine, Durham, NC, USA. [5]Department of Neurobiology, Duke School of Medicine, Durham, NC, USA. [6]Penn Epilepsy Center, Perelman School of Medicine, University of Pennsylvania, Philadelphia, PA, USA. [7]Department of Neurology, Duke School of Medicine, Durham, NC, USA. [8]Department of Psychology and Neuroscience, Duke University, Durham, NC, USA. [9]Center for Cognitive Neuroscience, Duke University, Durham, NC, USA. ✉e-mail: j.viventi@duke.edu; gregory.cogan@duke.edu

into words[19,20], and sentences[21,22]. Accurate resolution of these features is therefore a crucial component for speech decoding.

The rich spatio-temporal structure of human brain signals occurs at small spatial scales. Recordings of brain signals have previously been limited by ECoG recordings, which are typically measured from 64 – 128 contacts spaced 4 – 10 mm apart, or SEEG recordings that use depth probes (8 – 16 contacts) to measure cortical signals at 3.5 – 5 mm spatial resolution. This limitation is particularly relevant for signals that are both spatially specific and highly informative. One such signal is the high gamma band (HG: 70 – 150 Hz), which has been shown to index local neural activity from the surface of the brain[23]. HG has a high correlation with multi-unit firing and also shows high spatial specificity[24,25]. Further, HG has been previously shown to more accurately estimate neural-firing patterns that are stable over longer periods as compared to single units[26]. Micro-scale neural recording of HG could therefore enable accurate resolution of speech-articulatory features.

Information across even small distances in the human brain is distinct during speech production. Previous methods that quantified HG signal-sharing in SMC during speech articulation have shown low inter-electrode correlation (r = 0.1-0.3 at 4 mm spacing), indicating that articulatory neural properties are distinct at millimeter scale resolutions[27–29]. Consequently, speech decoding studies that utilized HG activity have identified a boost in performance with higher-density cortical sampling[20,21]. Neural speech decoding using 4-mm-spaced arrays showed up to a 5× increase in phoneme prediction compared to 10-mm-spaced arrays[15,18]. These results show that decoding performance improved with increased electrode density, further motivating higher resolution neural recordings to accurately resolve HG for reliable speech decoding.

Previous work using high-density micro-electrocorticographic (μECoG) arrays in other domains have shown the ability to resolve micro-scale neural features. In rodents and non-human primates, sub-millimeter spacing revealed fine-scale sensory topologies consistent with intracortical electrodes[30–33]. In humans, this high-resolution electrode technology has also enabled the identification of micro-scale epileptic signatures of seizure-onset-zones in epileptic patients[34–38] and have shown improvements in motor neural prostheses by resolving the motor cortex at millimeter-level resolution[39–42]. These results show that high-resolution neural recordings could resolve micro-scale articulatory features from SMC.

In the present work, we demonstrate the use of high-density μECoG for speech decoding in humans. We recorded intra-operatively from speech-abled patients using liquid crystal polymer thin-film (LCP-TF) μECoG arrays (1.33 – 1.72 mm inter-electrode distance, 200 μm exposed diameter electrodes) placed over SMC during a speech production task. These recording devices enable the high-resolution spatio-temporal sampling of local neuronal activity which produced superior speech decoding. We decoded speech by predicting the actual spoken phonemes from HG neural activations. We compared our results from high-density μECoG to neural decoding from standard intracranial electroencephalographic (IEEG) recordings, to empirically validate our improved decoding results. We also show that high-density μECoG decoding relies on the ability to resolve micro-scale spatial and temporal features of the neural signal. Lastly, we leverage this access to micro-scale neural signals to enable a nonlinear decoding model to decode entire speech sequences. We show the use of high-density μECoG for neural decoding of speech. This high-spatial sampling technology could lead to improved neural speech prostheses.

## Results
### High resolution neural activity of speech
We examined micro-scale speech neural activity from four subjects (1 female, mean age = 53) recorded using LCP-TF μECoG electrodes in the intraoperative setting. We used two versions of the LCP-TF μECoG electrode arrays to record speech neural activations from

SMC: a 128-channel subdural array (Fig. 1a, b **(top)** 8 × 16 array; inter-electrode distance: 1.33 mm) and a 256-channel subdural array (Fig. 1a, b **(bottom)** 12 × 22 array; inter-electrode distance: 1.72 mm). The μECoG electrode arrays had up to 57× electrode density with respect to macro-ECoG arrays and up to 9× higher density compared to high-density ECoG arrays (Fig. 1c). Subjects S1, S2, and S3 underwent surgery for the treatment of movement disorders and were implanted with a 128-channel array that was tunneled through the burr-hole during deep brain stimulator (DBS) implantation (see Methods). Each of the subjects completed three task blocks (52 unique tokens per block; three repetitions overall) of a speech repetition task, during which the subjects were asked to listen to and repeat back auditorily presented non-words. Each non-word stimulus was either a CVC or VCV token, with a fixed set of 9 phonemes (4 vowels and 5 consonants) at each position within the token (see Methods). Subject S4 underwent surgery for tumor resection and was implanted with a 256-channel array. This subject completed one block of the same speech repetition task as other subjects (Fig. 1d, Supplementary Fig. 2, Supplementary Table 1 & 2). Subjects took on average 1.1 seconds (range = 0.7 to 1.5 s) to repeat auditorily presented non-words and had an average spoken utterance duration of 450 ms (range = 300 to 700 ms) (Fig. 1e, Supplementary Table 3). Subjects correctly repeated the non-words on more than 95% of the trials (Fig. 1f, S1: 96%, S2: 98%, S3: 98%, S4: 100%), indicating that subjects understood and could complete the task in the intraoperative setting. The overall experiment time lasted up to 15 minutes and total utterance duration lasted 0.47 to 1.5 minutes (Supplementary Table 3).

We observed uniform in vivo impedance across the array (S1: 81.3 ± 3.8 kOhm, S2: 12.9 ± 1.3 kOhm, S3: 27.5 ± 4.8 kOhm, S4: 19.7 ± 4.2 kOhm, mean ± standard deviation; Supplementary Fig. 1 & 3), and discarded electrodes with higher impedance (>1 MOhm) from neural analysis. To confirm the absence of acoustic contamination in the neural data, we objectively examined the recordings for microphone contamination and did not observe a significant presence of microphone signals in our neural band of interest, across all subjects (except S1 at higher frequencies greater than 200 Hz; Supplementary Fig. 3). On examining speech neural activations, we observed significant modulation of spectro-temporal neural activations (multi-taper spectrogram estimate averaged across all spoken trials, see Methods) during speech articulation, including prominent HG band power increases. These distinct spatial patterns were seen in each patient and are shown in example arrays for 128 channel (S1) and 256 channel (S4) arrays in Fig. 2a. HG power increases were aligned to the speech utterance in individual electrodes and were active up to 1000 ms before utterance onset, Fig. 2b, Supplementary Fig. 4) and were identified as statistically significant as compared to a pre-stimulus baseline using a non-parametric permutation test with an FDR-corrected alpha threshold of p < 0.05 (see Methods). Significant electrodes are highlighted in Fig. 2 with black borders: S1 111/128 significant channels, S2 111/128, S3 63/128, and S4 149/256. These electrodes exhibited spatially varying characteristics of HG activations with 77.4% of these electrodes (S1 107/111, S2 107/111, S3 34/63, and S4 88/149) were active before the utterance start, indicating that μECoG electrodes measured earlier motoric activations leading to speech (Supplementary Fig. 5).

Next, we sought to determine the benefit of μECoG electrode for recording neural signals at higher fidelity as compared to standard methods. We examined the evoked-signal-to-noise ratio (ESNR) of HG power from μECoG neural recordings (−500 ms to 500 ms with respect to speech utterance onset) and compared it to standard IEEG (see Methods). We pooled within-subject IEEG electrodes (electrodes implanted during clinical pre-operative epileptic monitoring on separate patients: ECoG and SEEG) that were anatomically localized to SMC (Supplementary Fig. 14), and which exhibited significant HG power during speech articulation. Neural signals from μECoG recordings demonstrated a 48% (1.7 dB) increase in measured signal-to-noise ratio

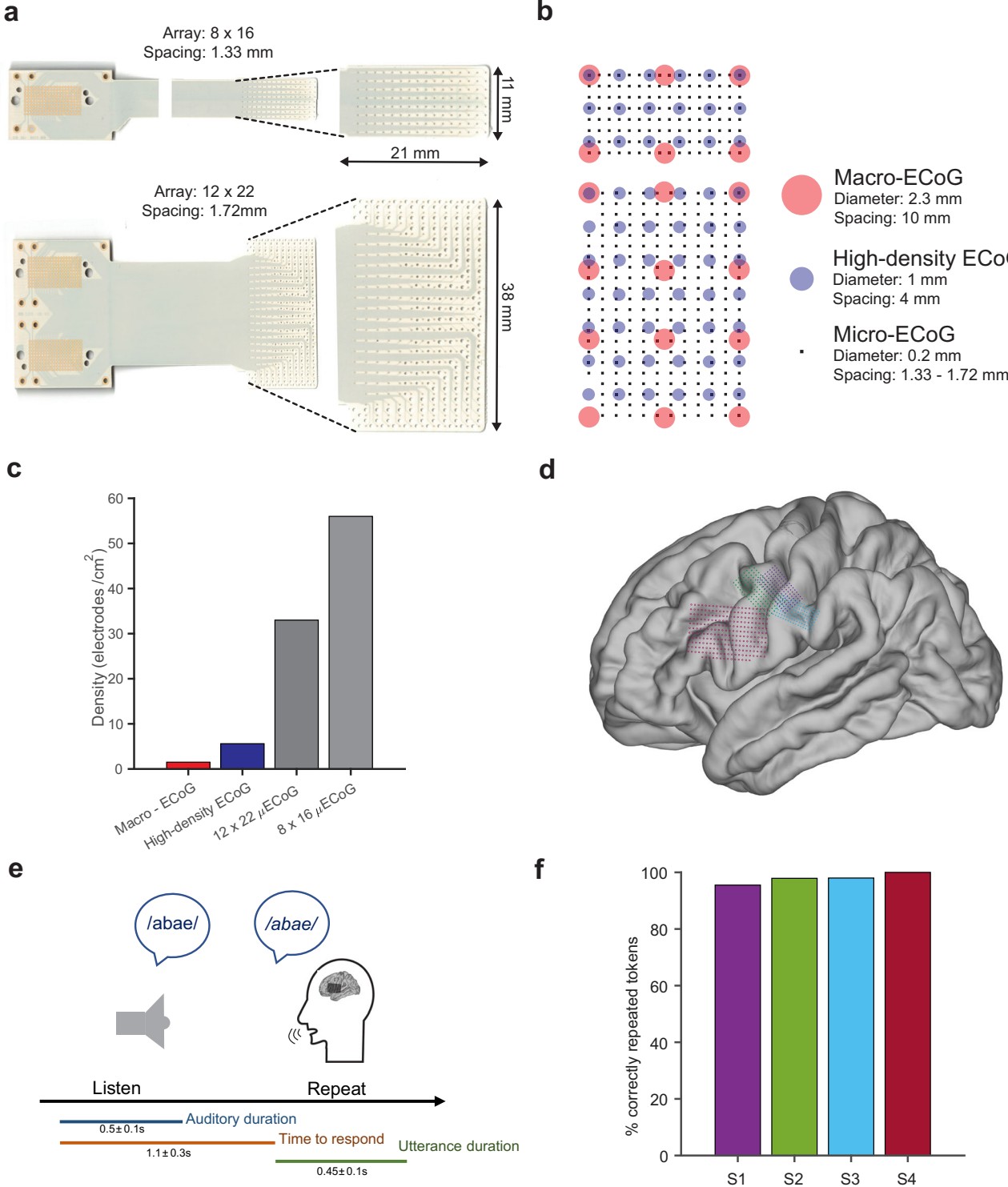

**Fig. 1 | Recording from high-density micro-electrocorticographic (μECoG) electrodes on the human brain during a speech production task. a** Flexible liquid crystal polymer (LCP) electrode arrays with 128 (top) and 256 (bottom) recording electrodes (200-μm diameter). **b** Visual comparison of high-density micro-electrodes (black square markers) with macro-ECoG (red circles) and high-density standard ECoG (blue circles). **c** μECoG electrodes exhibit higher spatial density (34 – 57 electrodes/cm²) compared to the existing macro-ECoG (1 electrode/cm²) and high-density ECoG arrays (6 electrodes/cm²). **d** Electrode arrays were implanted over speech motor cortex (SMC) in four awake patients (projected onto an average MNI brain). Electrode arrays had either 128 channels with 1.33 mm center to center spacing (pitch) for S1 (violet), S2 (green), and S3 (blue), or 256 channels with 1.72 mm pitch for S4 (red). **e** A schematic of the intraoperative speech production task. Color bars indicate the duration of the auditory stimulus (blue), time-to-response (orange), and spoken duration (green). **f** Subjects performed the speech production task with behavioral results above 95% accuracy for correctly repeated non-words. Source data are provided as a Source Data file.

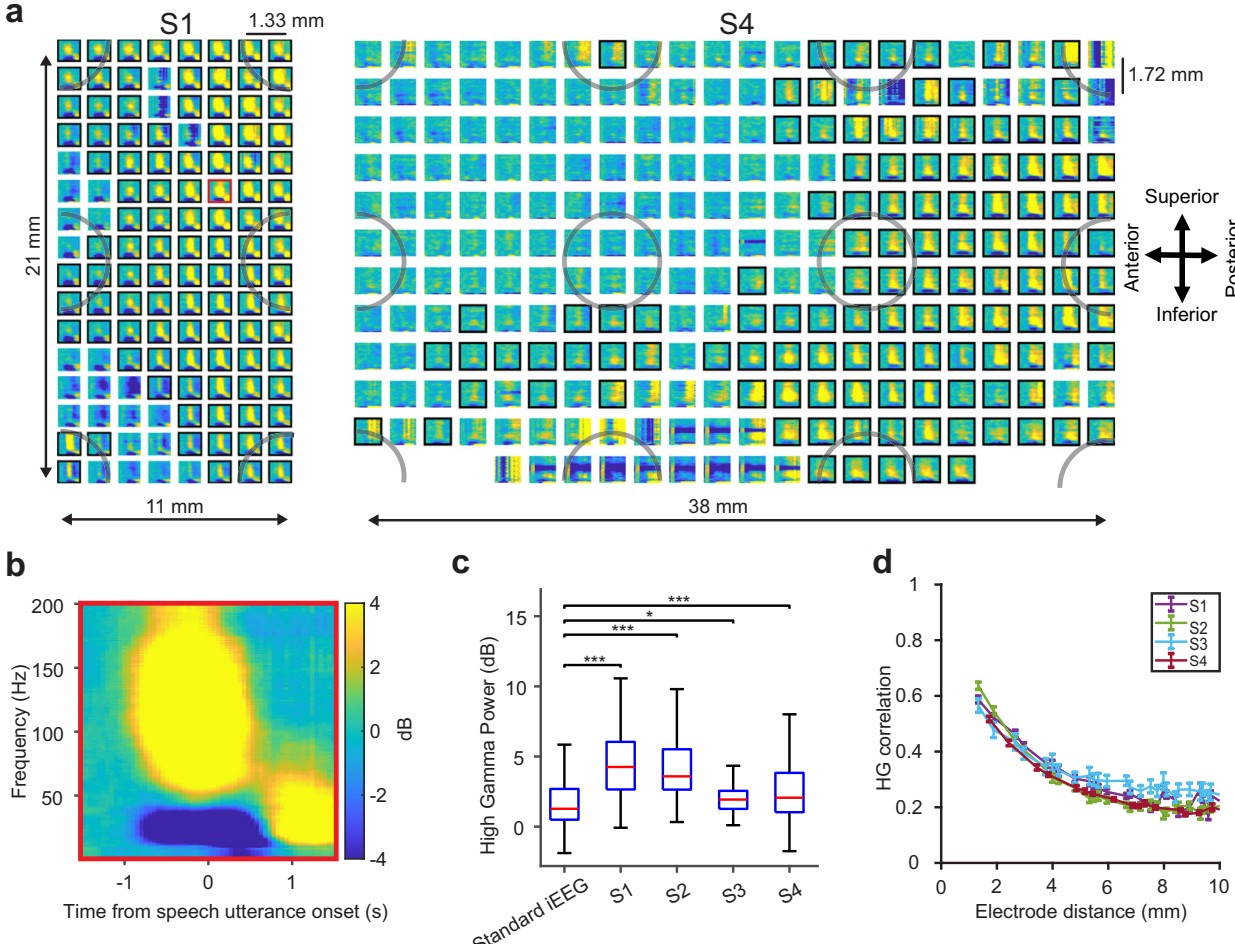

**Fig. 2 | Highly resolved neural features as demonstrated by high-density μECoG during speech production. a** Spectrograms for each electrode from each array for two example subjects (averaged across all spoken trials): S1, 128 electrodes (1.33 mm spacing) & S4, 256 electrodes (1.72 mm spacing). Data for all subjects are available in the supplemental materials. Electrodes with significant high-gamma modulation (HG: 70 – 150 Hz, FDR-corrected non-parametric permutation tests) during the speech utterance are indicated by black borders. Shaded grey circles represent superimposed simulated macro-ECoG electrode resolution with 10 mm interelectrode distance, illustrating the increased density of μECoG as compared to standard IEEG. **b** A spectrogram of an example electrode from the 128-channel array (red border) demonstrated increase power in HG band which was time-locked to the speech utterance. **c** μECoG arrays captured HG activity at significantly higher power than standard IEEG (ECoG and SEEG), *$p < 0.05$, ***$p < 0.001$, one-sided

Mann-Whitney U test, showing the increased ability of micro-electrodes to measure neural signals with a higher signal-to-noise ratio as compared to standard methods (IEEG vs. S1: $p = 1.4e\text{-}14$, IEEG vs. S2: $p = 1.7e\text{-}5$, IEEG vs. S3: $p = 1e\text{-}3$, IEEG vs. S4: $p = 3.3e\text{-}5$). The red lines and the blue boxes indicate the median and 25th/75th percentile and solid lines represent the full range of the distribution (total number of electrodes in each grouping, standard IEEG: 60, S1: 111, S2: 111, S3: 63, S4: 149). **d** Inter-electrode HG correlation decreased with increased electrode spacing for all subjects. The correlation values at each electrode spacing are represented by mean and standard error ($n$ = all possible electrode pairs at fixed electrode distance in mm). The correlation values at spatial resolutions less than 2 mm (r = 0.6), reveals evidence for spatially discriminative neural signals at the micro-scale during a speech utterance. Source data are provided as a Source Data file.

(median HG ESNR = 2.9 dB, S1: 4.2 dB, S2: 3.5 dB, S3: 1.9 dB, and S4: 2.1 dB, $p < 0.05$, Mann-Whitney U test, Supplementary Fig. 6) as compared to recordings from standard IEEG (median ESNR = 1.2 dB, Fig. 2c). These neural HG activations exhibited fine-scale spatial tuning across the array, indicating that speech informative electrodes can be spatially clustered (Supplementary Fig. 6).

To quantify the spatial resolution of μECoG neural signals, we computed the Pearson correlation coefficient of the HG envelope[27] (−500 ms to 500 ms with respect to speech utterance onset) between each micro-electrode pair. The correlation increased with proximity from 10 mm to 1.33 mm and remained largely uncorrelated at the equivalent spatial resolutions of macro ECoG for all 4 subjects (Fig. 2d: r < 0.2 at 10 mm spacing). HG neural signals at less than 2 mm were more correlated (r = 0.6), but not fully correlated, suggesting that speech information contained in HG neural activations are spatially discriminative at scales below 2 mm.

To investigate micro-scale neural information, we examined spatio-temporal activations specific to key units of speech production: articulatory features of individual phonemes (low-vowel - /a/, /ae/, high-vowel - /i/, /u/, labial-consonant - /b/, /p/, /v/, and dorsal tongue consonant - /g/, /k/). We first examined spatio-temporal activity patterns by averaging normalized HG activity across trials specific to first-position phonemes of the non-word utterances. All four subjects exhibited distinct spatio-temporal patterns for four different articulators (/a/ - low, /i/ - high-vowel, /b/ - labial, /g/ - dorsal tongue) with respect to the speech utterance onset (Fig. 3 & Supplementary Fig. 7), as highlighted in Fig. 3a for example subject S1.

We next sought to investigate how motor articulatory features were organized in neural space. Previous studies examining speech production have identified functional organization patterns of phonemes based on their composite articulators in SMC[43,44]. We sought to identify similar population-driven articulator patterns for μECoG

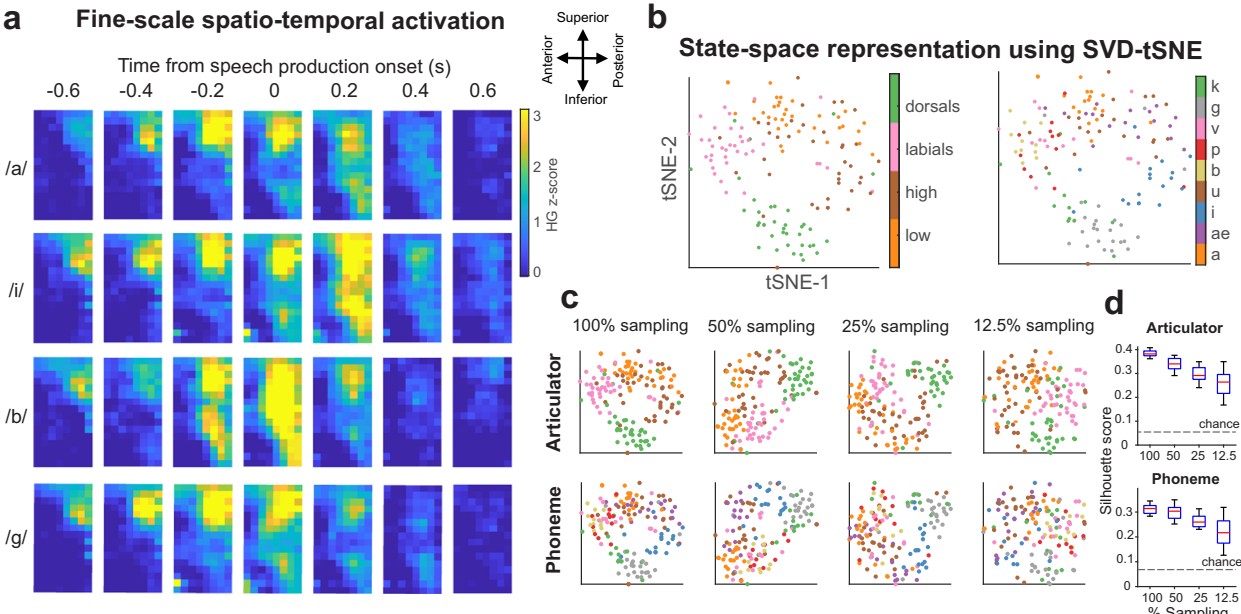

**Fig. 3 | μECoG captures fine-scale spatio-temporal activity during speech production. a** HG activations showed fine-scale spatio-temporal patterns in SMC, shown here for an example subject (S1) during speech production. Spatial maps of HG responses showed discriminative activation patterns for each phoneme. **b** State-space analysis using a signal value decomposition t-distributed stochastic neighbor embedding method (SVD-tSNE) depicted clear hierarchical separation of trials with respect to articulators (dorsal - /g, k/, labial (/b, p, v/), high (/i, u/), and low (/ae, a/) as well as individual phonemes. Phonemes produced using similar articulators were grouped together in two-dimensional state-space suggesting similarity in motor activation patterns. **c** Electrode sub-sampling to simulate lower resolution recordings reduced the separation of clusters for both phoneme (top) and articulator (bottom) groupings in SVD-tSNE state-space (same color palette as in (**b**)). This reduction of grouping demonstrates the utility of high-resolution spatial sampling. **d** To quantify this reduction, we performed a silhouette analysis that measures the relative closeness of a trial to articulator/phoneme group with respect to other groups. To show this significant decrease in cluster separation/ groupings empirically, we performed a one-way ANOVA (articulator: $F_{3, 196} = 114$, $p = 9.6e\text{-}43$, phoneme: $F_{3, 196} = 47.8$, $p = 3e\text{-}23$) and post hoc $t$-tests (see main text). Each box plot depicts the distribution of silhouette scores obtained from electrode sub-samplings (50 samples using Poisson disc sampling). The red lines and the blue boxes indicate the median and 25th/75th percentile and dashed lines represent the full-range of the distribution. The dotted horizonal line indicate chance silhouette metric obtained by shuffling the articulator/phoneme labels (1000 iterations). These analyses demonstrate the improved ability of high-resolution neural sampling to distinguish between articulator/phoneme motor features. Source data are provided as a Source Data file.

HG (500 ms window; 100 time points) in a cortical state-space and determine if this organization is dependent on spatial resolution. To model cortical state-space, we used singular value decomposition (SVD) to transform spatio-temporally covarying HG activity into low-dimensional features (80% of variance explained), and used t-distributed Stochastic Neighbor Embedding (tSNE)[45] to visualize the transformed principal-component scores in a two-dimensional vector space. Figure 3b shows clear separation of speech trials in tSNE state-space for both articulators as well as individual phonemes. We examined the contribution of the high spatial-resolution of μECoG by repeating the state-space analysis of spatially subsampled HG (Poisson disc sampling spatially constrained across the array, Supplementary Fig. 8a). We observed decreased distinction for both articulators and phonemes with reduced spatial sampling (Fig. 3c). This effect is further quantified by a significant decrease in silhouette scores for sub-sampled versions in both articulator and phoneme space (Fig. 3d, Supplementary Fig. 8b). To statistically assess this difference, we performed a one-way ANOVA to examine the effects of spatial subsampling on state-space clustering which revealed a significant main effect of subsampling in both articulator ($F_{3, 196} = 114$, $p = 9.6e\text{-}43$) and phoneme space ($F_{3, 196} = 47.8$, $p = 3e\text{-}23$). Post hoc $t$-tests with Bonferroni correction showed that mean values were significantly different between all groups in the articulatory space ($p < 0.01$). Differences were also significant in the phoneme space, except for between 100% and 50% subsampling ($p = 0.17$). These results clearly demonstrate that high-resolution neural recordings enable more accurate representation of motor features in SMC.

## Decoding phonemes using high-density μECoG spatio-temporal features

Successful neural speech prostheses will require the ability to decode the full range of human speech and language. One solution to enable this ability is to focus on compositional units that enable generative language. We therefore sought to evaluate the ability of high-density μECoG to decode the smallest compositional unit of speech: the phoneme. We performed a 20-fold, nested, cross-validated decoding model of manually aligned phonemes based on a low-dimensional subspace (SVD decomposed) using a supervised linear discriminant model (LDA; see Methods). We selected the eigenvalues based on nested cross-validation that explained 80% of neural variance (equivalent number of principal components: S1 − 34, S2 − 39, S3 − 25, S4 − 22) for phoneme prediction. Benefiting from high spatial sampling, we observed strong decoding performance in all subjects for predicting phonemes in all positions within the non-word (e.g., /abae/: P1 - /a/, P2 - /b/, P3 - /ae/; Fig. 4a, b, $p < 0.01$, Binomial test against chance model[46]). All subjects had higher accuracies for decoding phonemes in the first position (average P1 across subjects: 49.9%, chance = 11.11%) when compared to phonemes in the second (P2: 45%, $p = 0.0097$) and third positions (P3: 38.8%, $p = 0.0250$; one-sided paired $t$-test). This decoding performance was specific to HG information alone as adding low frequency signals (LFS) did not result in performance increases (Supplementary Fig. 9). S1 and S2 exhibited higher decoding performances across all positions which we attribute to higher HG-SNR and increased utterance duration for these subjects (Fig. 4b). S1-P1 (Subject 1, phoneme position 1) had a maximum prediction score of 57% with the best consonant (/g/) obtaining a decoding

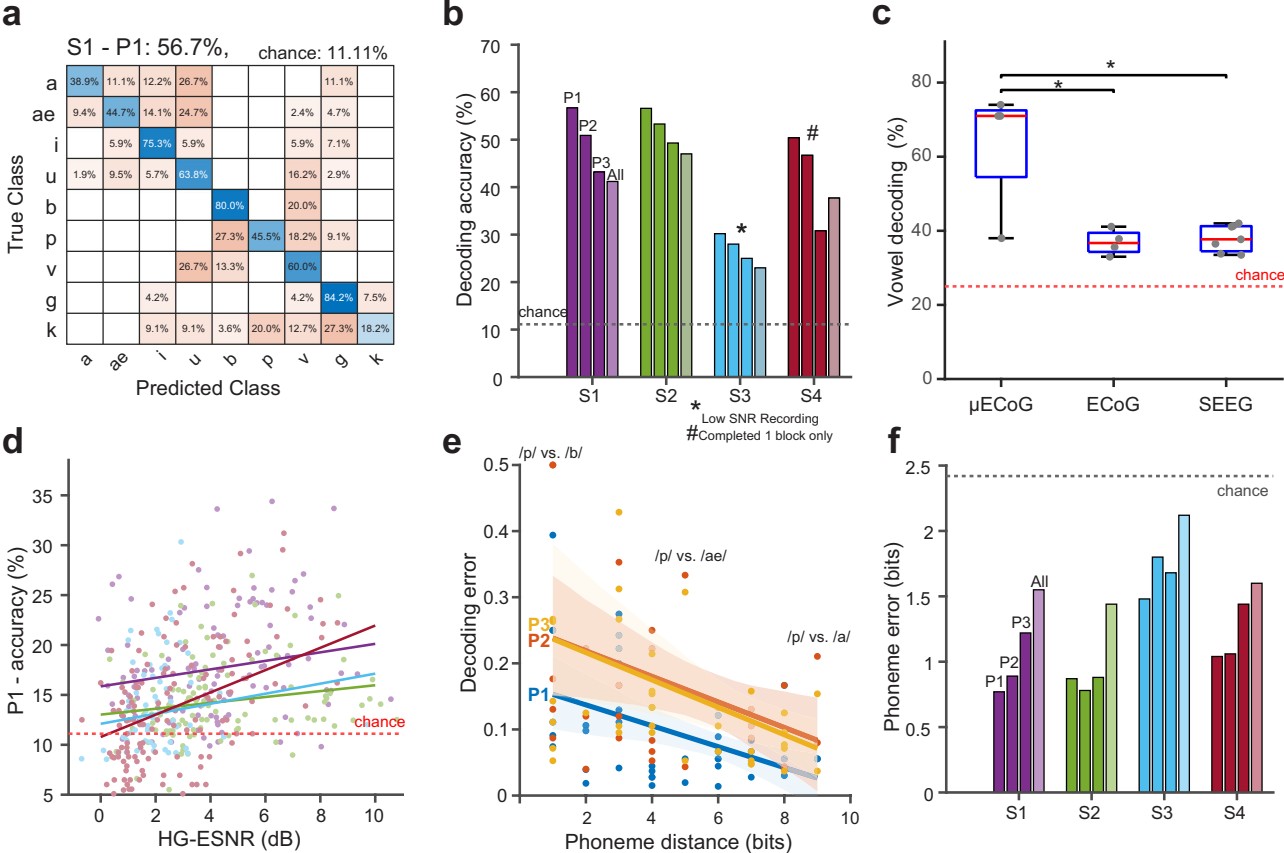

**Fig. 4 | High accuracy phoneme decoding from µECoG. a** Confusion matrix for an example subject (S1) for phoneme position one (P1) using singular value decomposition linear discriminant analysis (SVD-LDA) decoding demonstrated high accuracy for each phoneme. **b** Decoding was also highly accurate for each position within the non-word (P1, P2, P3), as well as for all phonemes combined across position (mean P1 = 49.9%, P2 = 45%, P3 = 38.8%, All = 36.6%, and chance = 11.11%). **c** This accurate decoding outperformed standard IEEG in a 4-way vowel decoder (P1: µECoG (n = 4): 71%, vs. ECoG (n = 4): 36.7%, p = 0.015, vs. SEEG (n = 7): 36.5%, p = 0.011; chance = 25%, one-sided permutation test with 10,000 iterations). This nearly doubled performance increase further demonstrates the benefit of higher resolution neural recordings for speech decoding. The red lines and the blue boxes indicate the median and 25th/75th percentile and dashed lines represent the full-range of the distribution. **d** Distribution of single-electrode decoding values were correlated with HG-ESNR for all subjects. Linear model fits indicated moderate but significant correlations for S1 (purple - Pearson r = 0.23, p = 0.0087), S2 (green - r = 0.23, p = 0.0077), and S3 (blue - r = 0.19, p = 0.0303), and a stronger correlation for S4 (red - r = 0.44, p = 6.5e-14, F-statistic test against constant model). **e** Decoding errors decreased with phoneme distance for all positions indicating higher confusion between phonemes with shared articulatory features (see methods and supplement for how error/phoneme distance was computed). Solid lines indicate linear model fit obtained for each position (P1: $F_{1,79}$ = 90.4, p = 9.9e-15, P2: $F_{1,79}$ = 19.7, p = 2.8e-5, P3: $F_{1,79}$ = 28.6, p = 8.4e-7; F-statistic test against constant model) and shaded regions indicate the 95% confidence interval of the linear model fit obtained using bootstrap regression (n = 1000 bootstraps). **f** Phoneme errors at all three positions in the non-words and all phonemes combined across position show decoding errors (bits) well below chance (chance = 2.4 bits for a uniform confusion matrix). Source data are provided as a Source Data file.

score of 84.2% and the best vowel (/i/) obtaining a decoding score of 75.3% (Fig. 4a). Similarly, S2-P1 had similar maximum prediction scores with the best consonant (/g/) with a decoding score of 72.7% and the best vowel (/u/) with a decoding score of 83.3%. Decoding accuracies were reduced when manually aligned phonemes were pooled across all positions (S1: 40.72%, S2: 42.96%, S3: 20.92%, and S4: 41.67%). This reduction in performance suggests that there is unique position-based information present in the signal, and decoding can be impaired by neural correlates from neighboring phonemes, therefore, indicating a need for position-dependent decoding. The lower prediction quality of S3 was attributed to poor contact between the electrodes and SMC (as indicated by low HG-SNR). S4 had comparable SNR as S1 and S2, however this subject completed only one out of three blocks during the intraoperative experiment resulting in reduced data duration for the decoding model and highly varying performance across phoneme positions. To show that the reduced performance obtained from S4 was likely due to this decreased recording duration, we subsampled trials across all subjects and phoneme positions and found comparable decoding accuracy when the recording duration for other subjects was

also reduced (Supplementary Fig. 10, median accuracy at 50 subsampled trials – S1: 36%, S2: 27%, S3: 15% and, S4: 44%). Although higher spatial coverage of S4 enabled recordings from both inferior frontal gyrus (Broca's area) and SMC, significant decoding was largely specific to electrodes over SMC (Supplementary Fig. 11). The decoding results used a fixed time-window (−500 ms to 500 ms) across phoneme onsets. To determine the optimal time-window required to decode phonemes, we calculated the decoding scores at varying time-windows for all subjects and phoneme positions (Supplementary Fig. 12). The decoding performance increased with longer windows until saturation at around 500 ms. This saturation point was specific for each subject and phoneme position, and the optimal decoding values were greater than or equal to values observed with fixed time-windows. S3 and S4 exhibited a reduction in decoding beyond the optimal point, indicating that performance could be impaired by adjacent phonemes at low SNR and reduced data duration. Our decoding results represent the best phoneme decoding scores for surface neural recordings and validate the use of high-density µECoG for the development of neural speech prostheses.

To directly assess the benefit of recording neural signals for speech at the micro-scale, we compared high-density µECoG decoding to standard IEEG recordings (SEEG/ECoG). We performed the same phoneme decoding analysis using neural signals recorded from eleven patients with epilepsy (7 females, mean age = 28.5, 4 ECoG, 7 SEEG) who were implanted with standard IEEG electrodes during pre-operative epilepsy monitoring. For comparison between electrodes, we performed a 4-way decoder (using SVD-LDA; see Methods) to decode between vowels and quantified the decoding performance. For each patient, we anatomically restricted electrodes to SMC and selected the electrodes that exhibited significant HG ESNR during the speech utterance (Supplementary Fig. 16). Our results showed that on average, µECoG outperformed standard clinical recordings by 36% (Fig. 4c: ECoG: 36.7%, SEEG: 36.5%, µECoG: 71%, chance 25%, $p < 0.05$, one-sided permutation tests with 10,000 iterations). This boost in performance results show the ability of high-density µECoG to enable more accurate speech decoding as compared to standard clinical recording techniques.

Since high-density µECoG demonstrated both a greater ability to resolve higher ESNR and stronger decoding performance as compared to standard clinical recordings, we next sought to directly determine the relationship between decoding performance and ESNR. To quantify this relationship, we calculated the univariate phoneme decoding performance for every individual µECoG electrode along with their corresponding HG-ESNR (dB) value (Fig. 4d). Linear model fits indicate moderate but significant correlations for S1 (Pearson r = 0.23, $p = 0.0087$), S2 (r = 0.23, $p = 0.0077$), and S3 (r = 0.19, $p = 0.0303$), and a strong correlation for S4 (r = 0.44, $p = 6.5e-14$). Further, we observed increases in accuracy as a function of HG-ESNR at a mean rate of 0.4% per dB for S1, S2, and S3, and 1.1 % per dB for S4. These results show a moderate but significant relationship between the ability to capture higher SNR micro-scale activations using high-density µECoG and the corresponding ability to decode speech.

Finally, we sought to characterize the types of decoding errors that were present in high-density µECoG recordings. We reasoned that if our ability to decode phonemes was largely based on our ability to resolve their constituent articulatory features in the motor cortex, decoding errors should not be uniform across phonemes, but rather reflect similar articulatory properties (low, high, labials, and dorsals) from other phonemes. For example, in Fig. 4a, we see increased confusion between phonemes with similar phonological features (bilabial plosives /p/ vs. /b/)[6,15,17]. To systemically characterize these types of errors, we calculated each phonemes' phonological feature set[47] by assigning a 17-bit categorical vector to each phoneme (see Methods). We used Hamming distance to calculate phonological distance for each phoneme pair (Supplementary Fig. 13) so that phonemes with similar phonological features had lower distances (e.g., /p/ & /b/ differed by 1 bit, whereas, /p/ & /g/ differed by 5 bits). We found that our decoding models' confusion/error rate decreased with phonological distance across all phoneme positions (Fig. 4e, P1: $F_{1,79}$ = 90.4, $p = 9.9e-15$, P2: $F_{1,79}$ = 19.7, $p = 2.8e-5$, P3: $F_{1,79}$ = 28.6, $p = 8.4e-7$; F-statistic test against constant model). We observed similar inverse relationships for all subjects and the errors were lower than chance, (Fig. 4f chance error: 2.42 bits; uniform confusion matrix). S1 and S2 had the lowest phonological distance error across all phoneme positions (less than 1 bit; except S1-P3), indicating that phoneme misclassifications were predominantly biased by similar phonological feature articulation. This analysis showed that decoding using high-density µECoG exhibited errors that were systemic, largely due to speech articulatory structure and not random misclassifications.

## Improved decoding with increased resolution, coverage, and contact size

We next sought to directly study the effect of spatial resolution and spatial coverage of neural signals and their effect on decoding. For our subsequent analyses, we focused on S1 and S2 as they both had the requisite SNR and amount of data for our decoding analyses (see Supplementary Figs. 6 & 10 for a direct comparison). First, we examined the effect of spatial resolution on decoding by sampling electrodes ($n \geq 50$ samples) across the array, using Poisson disc sampling (Supplementary Fig. 8a). For each sampling, the resultant spacing in millimeters was calculated using the formula:

$$\text{pitch} = \sqrt{\frac{X * Y}{n}} \qquad (1)$$

where $X$ and $Y$ are length and width of the array in mm, and $n$ is the total number of electrodes in the sample. Accuracy values from the resultant spacing (rounded to the nearest integer) were then compared to the 95% threshold of maximum decoding accuracy values obtained from the full array. We found a sharp increase in decoding performance with an increase in spatial resolution. The subset of electrodes with spacing less than 1.5 mm reached the 95% threshold of the full array (Fig. 5a left, & Supplementary Fig. 14a). The subsampling distributions from the 10 to 4 mm range (electrode distance range for standard IEEG) had median accuracy values of only 20-35% as compared to 55 − 60% for the full micro-array, an improvement from 17% to 29%. Next, to demonstrate the importance of spatial coverage, we subsampled electrodes from rectangular subgrids at various fixed spatial resolutions. Each of these subgrids were two dimensional windows with dimensions from 2 × 4 up to 8 × 16 electrodes. The decoding performance improved with increased coverage (Fig. 5a middle & Supplementary Fig. 14a). Finally, to assess the impact of micro-scale contact-size on speech decoding, we estimated various sized recording contact areas of recording by using rectangular subgrids with dimensions from 1 × 1 (effective 200 µm contact size) to 8 × 8 (effective 10 mm contact size) of spatially averaged raw neural recordings. HG signals extracted from these spatially averaged neural recordings were then used to decode phonemes. Decoding accuracy decreased with increased estimated contact size. The highest performance was at the smallest contact size (Fig. 5a − right & Supplementary Fig. 14a). We validated the robustness of this spatial analysis by showcasing similar decoding trends across all phoneme positions for both S1 and S2 (Supplementary Fig. 14b).

Since previous decoding work has characterized a widely different number of phonemes, we next sought to characterize this spatial degradation of different N-way decoding models. For each electrode subsampling, we averaged the decoding accuracies of all possible N-way classifiers and obtained the simulated spacing required to achieve 90% and 95% threshold of full array decoding performance. We found that the spacing required to obtain the full array decoding performance was inversely proportional to the number of unique phonemes, indicating that the benefit of using high resolution neural recordings increases with the number of uniquely decoded phonemes (Supplementary Fig. 14c), further suggesting that as models scale up in size, the importance of high spatial sampling will be even larger. Together, these results directly validate the benefits of high-density µECoG for highly accurate neural decoding through their high-spatial resolution, large spatial coverage, and micro-scale electrodes.

We next used our recordings of micro-scale signals to identify articulatory motor maps in SMC. These maps show the specific neural organization of effectors for motor speech output. Previous studies have identified average maps across multiple subjects in SMC for articulator encoding[44], however, subject-specific maps have not been well characterized due to the relatively sparse coverage using standard electrode arrays. Individual subject maps would better enable the development of subject-specific neural prostheses. To quantify these micro-scale neural articulatory maps, we trained an SVD-LDA model for each electrode to decode four articulatory features (low vowels, high vowels, labial consonants, and dorsal-tongue consonants).

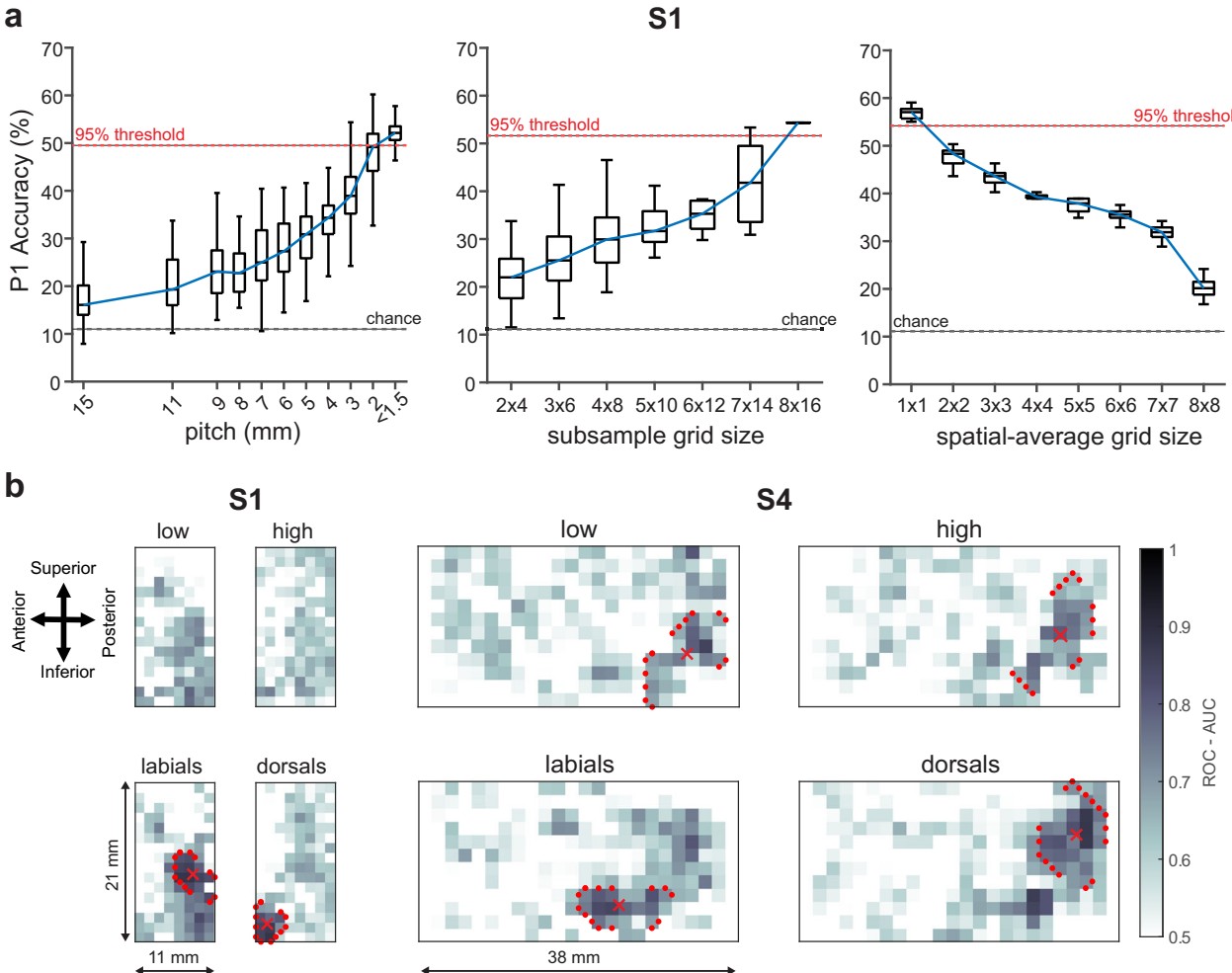

**Fig. 5 | Accurate decoding is dependent on the high-spatial resolution, high-channel count capability, and micro-scale recording of µECoG. a** Accuracy of P1 decoding increased as a function of spatial resolution and coverage as demonstrated by sub-sampling electrodes and decreased with contact size as demonstrated by spatial averaging. Each box plot shows the distribution of accuracy metrics from the subsampled electrodes (Electrode subsampling: $n \geq 50$ electrode subsamples obtained using Poisson Disk sampling, electrode subgrids: $n =$ all possible subsample grids defined by the coverage, spatial averaging: $n = 10$ instances of spatial averaged data). The horizontal center lines and the boxes indicate the median and 25th/75th percentile and vertical solid lines represent the full-range of the distribution. Blue lines connect the median accuracy values from the boxplots. Electrode pitch less than 2 mm along with micro-scale sampling was required to obtain decoding accuracy within 95% of the maximum decoding for the entire array (red dotted line) and demonstrated 29% improvement from the 10 mm pitch simulation and a 17% improvement from the 4 mm pitch simulation.
**b** Classification for articulators was spatially dependent, resulting in millimeter-scale spatially specific articulator maps shown here in example subjects S1 and S4. Darker colors show higher area under the curve for receiver operating characteristic curves (ROC-AUC) for the univariate linear decoding values for each articulatory feature. Performance for vowels (top) was spatially diffuse, whereas performance for consonant (bottom) formed spatially distinct clusters. The contour points identify 90th percentile of AUC values across all articulators, and the marker indicates corresponding centroids for each articulator. Source data are provided as a Source Data file.

The receiver-operator-characteristic area-under-the-curve (ROC-AUC) metric was used to evaluate each electrode's encoding preference for each articulator, and the values from all the electrodes were spatially arranged to derive micro-scale neural articulatory maps. We observed overlap of informative electrodes across each of the articulators. Despite this overlap, high-density µECoG arrays exhibited distinct clustering of articulatory features in all subjects, indicated by red contours (90 percentile of AUC) and corresponding centroids (Fig. 5b & Supplementary Fig. 15). For S1 with an 8 × 16 grid, AUC values were dispersed throughout the array for vowels, whereas there was distinct clustering for consonants (Fig. 5b, labial-consonants with 72.5 mm² surface area and dorsal-consonants with 17.7 mm² coverage, Supplementary Table 4). For subject S4 which had a 12 × 22 grid, we observed strong spatial clustering including both vowels and consonants (labials: 56.2 mm², dorsals: 79.9 mm², Supplementary Fig. 15 and Supplementary Table 4). These subject-specific, high-resolution articulator maps show the ability of high-density µECoG to resolve fine-scale

motor representations for speech production in individual subjects, which could enable better individualized neural speech prostheses.

## Sequential neural modeling reveals positional encoding of phonemes

Thus far our analyses have highlighted the ability of micro-scale recordings to resolve spatially specific neural signals. However, these signals reflect complex neural dynamics which also evolve in time. We sought to leverage our ability to resolve detailed spatial recordings of neural signals by examining how these signals evolve and change in time during speech utterances. We assessed these temporal dynamics through our ability to resolve phoneme sequences in time. We constructed an SVD-LDA model to train and test on HG neural time-segments with respect to the utterance. Sequences of the specific ordered phonemes within the utterance should only be resolvable if the sequence information is present in spatio-temporal HG neural activation patterns.

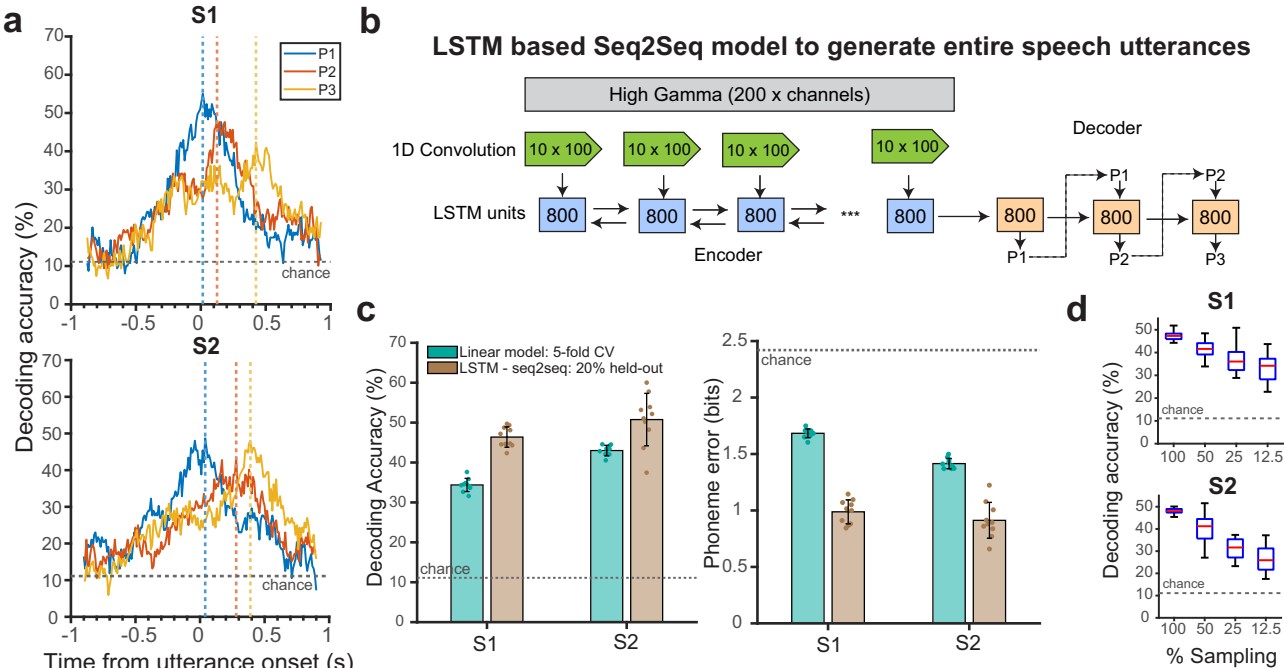

**Fig. 6 | µECoG enables the sequential temporal decoding of utterances. a** A temporal decoding model was trained on data aligned to the speech utterance onset for phonemes at each position. Results are shown for two example subjects (S1, S2) revealing temporal patterns of successful decoding for each phoneme position (P1: blue, P2: orange, P3: yellow). Dotted lines reflect peak decoding. This ability demonstrates that the temporal sequence of phoneme positions can be reconstructed via the information present in the neural signal aligned to the utterance. **b** We leveraged this ability by constructing a nonlinear decoding model to resolve temporal speech sequences of entire utterances. We used a long short-term memory (LSTM) Seq2Seq model with an encoder-decoder recurrent neural network (RNN) to translate HG spatio-temporal activation patterns (1 second window centered at utterance onset) to phoneme sequences, with three layers: a temporal convolution (green), a bi-directional encoder RNN (blue), and a decoder RNN (orange) layer. The numbers in the boxes represent the number of LSTM units (**c**) Our LSTM model outperformed a linear model for decoding phonemes across all positions, shown here for two example subjects, S1 (mean accuracy: 46.4% vs. 34.4%, mean phoneme error: 0.99 bits vs. 1.68 bits, $n = 10$ decoding instances on test trials) and S2 (accuracy: 50.8% vs. 43%, phoneme error: 0.91 bits vs. 1.4 bits, $n = 10$ decoding instances on test trials). Error bars indicate standard deviations, and the data points indicate performance metrics from each decoding instance on test trials. **d** Subsampling of electrodes ($n = 25$ instances of electrode subsamplings) reduced the LSTM performance in decoding phoneme sequences as validated by one-way ANOVA (S1: $F_{3, 99} = 40.47$, $p = 5.4e\text{-}17$, S2: $F_{3, 99} = 99.97$, $p = 2e\text{-}79$) and post-hoc $t$-tests (see Main text). The red lines and the blue boxes indicate the median and 25/75th percentile and dashed lines represent the full-range of the distribution. This improvement in decoding demonstrated the benefit of combining nonlinear decoding models that leverage high spatio-temporal neural activation with µECoG recordings that can sample the human brain at the micro-scale. Source data are provided as a Source Data file.

We obtained stable temporal decoding for phonemes across all positions. Further, the peak decoding values for each phoneme occurred sequentially times reflecting their order in the utterance (Fig. 6a: S1: 0.015, 0.125, and 0.425 s from the utterance onset for Position 1, 2, and 3 respectively, and for S2: 0.04, 0.28, and 0.39 s). This analysis demonstrates the ability to decode sequential phonemes in an utterance from the rich spatio-temporal neural features from micro-scale recordings. This ability to resolve these phoneme sequences also demonstrates our ability to resolve entire speech utterances from the decoding of phoneme units, a requirement for successful neural speech prostheses.

**Phoneme sequence generation using seq2seq recurrent neural network (RNN)**

Having demonstrated the presence of sequential phoneme information in the micro-scale neural signal, we sought to decode the entire utterance via the individual constituent phonemes without explicit phoneme onset information. We reasoned that this scenario would more accurately reflect the use-case of a neural speech prosthetic for entire utterances, where manual phoneme boundaries are not available. We developed a phoneme generation technique to sequentially decode positional phonemes within the utterance, without prior knowledge of the statistical relationship between phonemes (phonotactics). We used a seq2seq RNN model to translate HG neural activity to the entire phoneme sequence directly[21]. For each spoken utterance,

HG activations (1 second window; 200 time points) from all significant µECoG electrodes were passed as input into an 'encoder-decoder' recurrent neural-network (Fig. 6b). The model was executed in three stages: 1) The temporal convolution layers (green arrows) to extract low dimensional neural features, 2) The bidirectional encoder RNN (blue boxes) to encode neural features along the temporal-domain. 3) The decoder RNN (orange boxes) to transform encoded neural features to phoneme sequences.

We trained the entire network using Adam[48] gradient descent optimization (learning rate = 1e-3) for 800 epochs (number of visits of training data), with $\ell2$ regularization applied to all layers. The network hyperparameters were estimated using nested cross-validation (see Supplementary Table 5).

We quantified the decoding performance by calculating the percentage of correctly predicted phonemes for each position. The resultant decoding accuracies and errors were considerably better for seq2seq models for both S1 (mean accuracy: 46.4% vs. 34.4%, mean phoneme error: 0.91 bits vs. 1.4 bits) and S2 (accuracy: 50.8% vs. 43%, phoneme error: 0.91 bits vs. 1.4 bits) as compared to equivalent linear decoding models (Fig. 6c and Supplementary Table 6). This result demonstrates that non-linear sequential decoding can more accurately reconstruct spoken sequences than linear methods that require explicit phoneme onset times. To further validate the importance of micro-spatial sampling for these results, we performed an analysis that spatially subsampled our electrodes which resulted in a reduction of the

sequential decoding performance (Fig. 6d S1: $F_{3,99} = 40.47$, $p = 5.4e\text{-}17$, S2: $F_{3,99} = 99.97$, $p = 2e\text{-}79$). Post hoc $t$-tests with Bonferroni correction showed significant differences in mean across all subsampling groups for both subjects ($p < 0.01$). This sequential decoding method provides the basis for an algorithmic framework for the decoding of entire speech utterances using high-resolution neural recordings.

## Discussion

We demonstrate the benefits of using micro-scale neural recordings for speech decoding and as an important component of neural speech prostheses. We successfully performed intra-operative recordings from four speech-abled patients (3 movement disorder patients and 1 tumor patient) who completed a non-word speech repetition task. Even with the time-constrained experimental procedure (up to 15 minutes of recording duration/ 1.04 minutes of total spoken duration; Fig. 1e, Supplementary Table 3), our µECoG electrodes recorded neural signals in SMC that produced significantly higher HG-ESNR as compared to standard intracranial methods (ECoG/SEEG). Electrodes with HG activations spanned both the pre-central (motor) and post-central (sensory) gyrus across the SMC for S1 (Supplementary Fig. 2 & 4). Subject S2 and S3 had predominant coverage in the pre-central and post-central gyrus, respectively. Subject S4 had a 256-channel array that extended anteriorly from SMC to the inferior frontal gyrus, however, most HG activations were confined to SMC. Measured HG from SMC exhibited spatially varying temporal characteristics, with activations starting up to 1 s before utterance onset and lasting until speech offset (Supplementary Fig. 5). These early activations sites could indicate neural populations within SMC that transform auditory programs to motor speech[9,29]. Future studies will examine the neural characteristics of these sensory-motor integration sites using micro-scale recordings[8,9]. The HG activations were distinctive at the millimeter scale during the speech utterance, as validated by inter-electrode correlation of r ~= 0.6 (Fig. 2d) at <2 mm spacing, which both extends previous work with coarser spatial sampling and demonstrates the informative nature of signals within SMC at the millimeter scale[27,28]. This result is also in keeping with previous work that sampled microscale epileptic activity using high-density µECoG arrays which revealed a striking heterogeneity in the spatio-temporal dynamics of interictal events[49–51]. Importantly, in that work, micro-seizures were most often observed occurring on just one to two electrodes spaced 750 µm apart and were not visible on neighboring electrodes. The higher SNR combined with evidence for distinct neural patterns of activations at the micro-scale motivates the benefits of using micro-recording technologies for human speech decoding.

On examining the spatio-temporal activation of HG, we performed a cortical state-space analysis that resulted in clear clustering of articulatory features and phonemes, which is consistent with previous work[43]. We built on this finding by showing that this clustering was dependent on micro-scale spatial resolution as a decrease in sampling produced significantly worse clustering results. This inverse relationship shows the benefit of high-resolution neural recordings to accurately model speech cortical dynamics.

We performed speech decoding by classifying phonemes from all positions within the spoken non-word. Using non-word stimuli in the speech task enabled us to remove the effects of higher-order language processes and to decode phonemes and articulatory features to clearly assess the benefits of using high-density µECoG to decode speech features from neural recordings. Previous speech decoding studies with phoneme classification results employed electrode technologies ranging from standard ECoG[14–16,18–21] to penetrating intracortical arrays[17]. These studies involved subjects performing speech production tasks in an epilepsy monitoring unit, with total utterance durations (actual speaking time) of >~10 minutes. The corresponding neural recordings enabled speech decoding by classifying up to 39 English phonemes. With our initial experiments constrained to

15 minutes of intra-operative time, we developed a 9-way phoneme decoder and reported stronger accuracy metrics from intra-operative data with total spoken durations of ≤ 2.5 minutes (Supplementary Table 3). We observed robust decoding for classifying spoken phonemes across all subjects, with maximum accuracy reaching up to 57% (chance: 11.11%) in predicting both consonants and vowels, and with S1 and S2 achieving the highest decoding performances (Fig. 4). Subject S3 obtained lower decoding values likely due to poor contact with SMC (indicated by lower SNR) that could be due to issues related with tunneling of the electrode array through the burr hole. Subject S4 obtained decoding performance that was on par with S1 and S2. This subject, however, performed only one block of our speech repetition task. On examining the impact of recording duration, we observed increased decoding performance with more trials/time for training, as expected (Supplementary Fig. 10). Further, S4's decoding performance with a 256-channel array at 50 subsampled trials was higher than the other subjects with 128 channel arrays, indicating that improvements in decoding accuracy may be possible with increased channel counts. This improved decoding with coverage was specific to electrodes from SMC, and HG activations from inferior-frontal regions did not significantly contribute to the performance (Supplementary Fig. 11). The decoding scores were obtained from the HG band (70 to 150 Hz) and adding LFS as additional features did not improve the performance (Supplementary Fig. 9). LFS were associated with the temporal structure of speech (prosodics and syllabic information)[52–55], and the lack of performance improvements may be attributed to the stimuli design of single non-word speech tokens, compared to naturalistic stimuli with varying syllabic/prosodic information.

The decoding performance decreased for phonemes embedded within the non-word, most likely due to discrepancies from marking onset times of embedded phonemes and co-articulatory effects that arise during phoneme transitions. To investigate these effects, we examined the amount of time necessary to decode each phoneme (Supplementary Fig. 12). We observed increased decoding performance with larger windows until saturation at around 500 ms similar to previous results[15,17]. This optimal window was specific for each position indicating a need for decoding models that do not require explicit phoneme onsets. This decoding window selection, however, is essential for online systems with an optimal tradeoff between speed and accuracy[56]. Finally, we directly compared the decoding ability of µECoG electrodes to standard IEEG (ECoG/SEEG) electrodes located with SMC. SEEG shanks contained depth electrodes (3.5 – 5 mm inter-electrode spacing) and the ECoG arrays contained electrodes (10 mm inter-electrode spacing) that were arranged in a rectangular lattice. In both cases, the electrodes that were directly present within/over SMC were utilized for decoding. We showed that even when with using short time periods, all three recording technologies obtained above-chance decoding performance, and micro-scale neural recordings using high-density µECoG outperformed current standard technology (Fig. 4c, and Supplementary Fig. 16). Our work is limited by not providing a direct comparison of decoding performance against high-density ECoG (4-mm inter-electrode spacing) which has been previously been employed to achieve high-quality naturalistic speech synthesis[57] and text generation[22]. Due to this limitation, we performed a subsampling analysis to simulate recordings at different spatial resolutions for a parametric and direction comparison. Future work will focus on using longer recordings, more naturalistic speech, and direct reconstruction of utterances from micro-scale neural recordings.

When we analyzed decoder performance, we found that the classification errors varied systematically as a function of the utterance. Phonemes with similar phonological features (e.g. /p/ and /b/) were more often confused compared to dissimilar phonemes that shared fewer features (/p/ and /g/)[15,17]. We quantified this phonological error using Hamming distance (bits) and showed that classifier errors were inversely proportional to phonological distance. Evaluating

speech decoding models with phonological distance could lead to error correction that comparatively weighs similar phonemes with a lower cost versus phonemes with dissimilar features[58]. This phonological error metric could be used in addition to classification accuracy to optimize cost-functions for speech decoding algorithms.

We characterized the importance of spatial characteristics of micro-scale neural dynamics for speech decoding. We observed improved phoneme decoding performance that increased with spatial resolution, spatial coverage, and decreased contact size. There was up to 29% improvement in decoding performance as compared to simulated spatial resolutions of standard ECoG recordings (4 – 10 mm; Supplementary Fig. 14). We also found that spacing was inversely proportional to the number of unique phonemes, suggesting that with the benefits of μECoG recordings would be even greater for larger datasets with a greater number of phonemes (Supplementary Fig. 14c). Higher spatial resolution was, therefore, required to achieve better decoding as more unique phonemes were included in the analysis. On a subject-specific scale, we observed high-resolution speech articulator maps, similar to previously established group-level organization[44] (Fig. 5b and Supplementary Fig. 15). We observed that the surface area for each articulator map ranged from 11 mm² to 80 mm² depending on the location and coverage of each μECoG grid (Supplementary Table 4). Overlaying these spatial clusters on each subject's individual brain, we made the following anatomical observations: S1 and S3 exhibited labial clusters near the posterior region of post-central (sensory) gyrus, and dorsal-tongue clusters (only S1) near the anterior-inferior region. S2 displayed labial and dorsal-tongue clusters over the pre-central (motor) gyrus; and S4 demonstrated distinct articulatory clusters within pre-central gyrus, indicating that electrodes with high speech articulator information were contained within SMC. This anatomical clustering that we show is similar to previously identified somatotopic representations using high-density ECoG grids (4 mm inter-electrode spacing), that revealed a posterior clustering of lip articulators and an anterior clustering of tongue articulators with respect to central sulcus[44]. Therefore, our high-density μECoG electrodes were able to maximally resolve these spatially confined articulatory 'hotspots', thereby, resulting in highly accurate decoding at less than 2 mm spatial resolution. While this spatial characterization resulted in anatomical maps on subject-specific brains, the individual electrode locations were obtained by manually overlaying the array templates on the markers from intra-operative imaging data. Future work will include designing μECoG grids with fiducial markers and developing automated co-registration techniques to more accurately localize μECoG array onto the individual subject's brain surface[59,60]. Nevertheless, these subject-specific articulator maps indicate the potential of high-density μECoG for future neural prostheses for individual patients.

Speech production involves the temporal sequencing of articulatory movements[43,44], and consequently the neural properties of articulators should be sequentially encoded. With our micro-scale neural recordings, we demonstrated sequential dynamics of articulation by successfully decoding individual phonemes in time. This temporal decoding reflected the correct order of serial phoneme execution during the speech utterance. These decoding activation patterns overlapped in time for each successive phoneme which could be evidence for co-articulatory dynamics during the speech utterance[43,61]. We then leveraged this ability to decode the temporal dynamics of the speech utterance to decode the entire utterance by using a deep-neural-net based sequential model that was able to decode phoneme sequences in the non-words with only 2 minutes of training data (Supplementary Table 3). We found a significant improvement when compared to our linear model that required explicit phoneme onset information. The sequential decoding performance was dependent on high resolution sampling, further highlighting the benefits of μECoG recordings. Since the speech production system involves continuous generation of motor articulatory features (position and trajectories of

the vocal tract system), our decoding using RNN can be further improved by using these articulatory features as intermediary, to subsequently decode phoneme sequences[57]. Decoding using deep-neural network requires a high number of parameters[17,18,62], and future work will study the relationship between high-fidelity cortical recordings and deep network complexity for successful speech decoding. These results are dependent on micro-neural activations without the addition of an explicit language model[19,22] to boost performance, thereby highlighting the role of the higher resolution of neural recordings from high-density μECoG for entire speech utterance decoding without strong priors. Future work, however, will explore this use of state-of-the-art, deep-learning-based, speech-reconstruction models with large vocabulary language models.

Our results highlighted the benefits of high-density, high-channel-count LCP-TF μECoG arrays for human speech decoding. Micro-scale recordings demonstrated more accurate speech decoding than standard IEEG electrodes and better resolved the rich spatio-temporal dynamics of the neural underpinnings of speech production. These findings motivate the use of high-density μECoG for brain-computer-interfaces to restore speech for patients suffering from debilitating motor disorders who have lost verbal communication abilities[22,63].

## Methods

All our research studies were approved by the Institutional Review Board of the Duke University Health System under the following protocol IDs:

Pro00072892: Studying human cognition and neurological disorders using μECoG electrodes Pro00065476: Network dynamics of human cortical processing.

### Participants

We recorded neural activity from speech motor cortex (SMC) of four awake patients (mean age: 53, 1 female patient) with no speech-impairments using LCP-TF μECoG arrays, during intraoperative procedures at the Duke University Medical Center (DUMC - Supplementary Table 1). Three of the four subjects (S1, S2, and S3) were being treated for movement disorders and the intraoperative recordings were performed during awake deep brain stimulator (DBS) implantation surgery for movement disorder treatment. The fourth subject (S4) underwent brain tumor resection and neural data were recorded during the awake craniotomy prior to resection. As a control, we recorded from eleven patients (mean age: 30, 7 female patients) with epilepsy (no speech-impairment) implanted with conventional electrodes (ECoG or SEEG) and recorded with a clinical IEEG system (Natus Quantum), during preoperative intracranial monitoring at DUMC. All subjects were fluent in English, and we obtained written informed consent in accordance with the Institutional Review Board at the DUMC.

### Intra-operative μECoG array implantation

We performed intraoperative recordings using non-commercial, custom designed LCP-TF μECoG arrays that were fabricated by Dyconex Micro Systems Technology (Bassersdorf, Switzerland) with two different designs. For the first design, we used an array with 128 micro-electrodes (200 μm electrode diameter) in an 8 × 16 design with 1.33 mm center-to-center spacing (Fig. 1a, b, top).

The 128-channel array design is narrow and flexible enough to be implanted through the burr-hole opened during DBS implantation. Each subject's cortical anatomy was assessed using preoperative MR imaging with contrast, to ensure the absence of large cortical bridging veins. Following the burr hole craniotomy, the surgeon tunneled the electrode array in the subdural space using neuro-navigation (Stealth Station Surgical Navigation, Medtronic, Minnesota) and preoperatively determined the cortical target (SMC). The tail of the array was then reflected such that the burr hole remained open for the placement of the DBS

electrodes. A clinically-indicated intraoperative CT was obtained to ensure accurate placement of DBS electrodes which was also used to localize the placement of the array. The DBS insertion canula was used as the ground/reference.

The second LCP-TF μECoG design had 256 microelectrodes (200 μm diameter) in a 12 × 22 array (1.72 mm spacing) and was used to target anatomy directly over eloquent cortex exposed during an awake craniotomy (Fig. 1a, b, bottom). Following clinically indicated cortical mapping, the array was placed over the functionally relevant cortex (SMC) and affixed to a modified surgical arm. A grounding/reference wire was connected to a surgical retractor attached to the subject's scalp. Finally, the corners of the array were registered using Brainlab Neuronavigation (Munich, Germany), and were colocalized to the subject-specific anatomical T1 structural image.

The arrays were connected to custom headstages with Intan Technologies RHD chips for amplification and digitization. These headstages were connected to an Intan Technologies RHD recording controller through micro- high-definition multimedia interface (μHDMI) cables passed outside of the sterile zone. The entire μECoG set up was sterilized using ethylene oxide gas at DUMC prior to surgical implantation.

To co-register the electrode arrays on the individual subject's brain, the array locations were first assessed using intraoperative CT scan (S1, S2, and S3) or registration markings from Brainlab Neuronavigation software (S4). For each subject, the cortical surface was reconstructed from a preoperative MRI image, using Freesurfer[64,65]. For subjects S1, S2, and S3, the array landmarks (distal ends) were then localized using BioImage Suite, after aligning the reconstructed T1 volume with CT scans. For subject S4, we used key anatomical landmarks to localize the BrainLab coordinates on the individual subject's reconstructed brain. Finally, to localize individual electrodes, the electrode array templates (for both 128 & 256 channels) were then mapped to each individual subject brain by manually aligning the template to the array locations. Figure 1d and Supplementary Fig. 2 show three templates of 128-channel grids (violet, green, and blue for S1, S2, and S3 respectively), and a 256-channel template (S4: purple) implanted over SMC of a subject averaged brain and subject-specific brains, respectively.

## Neural Recording

Neural data from the μECoG arrays were recorded with an Intan RHD recording system (using Intan RHX data acquisition software, v3.0; Intan Technologies, Inc.). The recordings were analog filtered between 0.1 Hz and 7500 Hz and digitized at 20,000 samples per second. The control subjects implanted with the standard IEEG system (ECoG or SEEG) during pre-operative epileptic monitoring. For ECoG monitoring, the subjects had Ad-Tech electrode grid (48 or 64 channels; Ad-Tech Medical Instrument Corporation) with 10 mm inter-electrode spacing and 2.3 mm exposed diameter. The subjects with SEEG monitoring were implanted with either PMT (PMT Corporation) or Ad-Tech depth electrodes with 3.5 mm and 5 mm inter-electrode spacing, respectively (Supplementary Table 1). The intracranial IEEG data were recorded using a Natus Quantum LTM amplifier (with Natus Neuroworks EEG software from Natus Medical, Inc.), with an analog filter between 0.01 Hz and 1000 Hz and digitized at 2,048 samples per second.

## Speech task

Each subject performed a speech production task during their intra-operative procedure. In each trial, they listened to and repeated one of 52 non-words that were designed to sample the phonotactics of American English (Supplementary Table 2). The non-words contained an equal number of consonant-vowel-consonant (CVC: 26) and vowel-consonant-vowel (VCV: 26) tokens, which included a total of nine different phonemes (four vowels: /a/, /ae/, /i/, /u/ and five consonants: /b/, /p/, /v/, /g/, /k/). Each trial lasted between 3.5 – 4.5 s including the

auditory stimulus (mean stimulus duration: 0.5 second) followed by a three second response window with a 250 ms jitter between trials. The trials were presented in three consecutive blocks with 52 unique non-word tokens shuffled for each block. The overall task time took approximately 15 minutes for each subject. The control subjects implanted with ECoG/SEEG during preoperative clinical monitoring in the epilepsy monitoring unit at the DUMC performed a similar speech production task, but some repeated different speech tokens. Equivalent phonemes and number of trials were used for the direct comparison between recording techniques. The speech task was designed using Psychtoolbox scripts in MATLAB R2014a.

## Audio recording

Auditory stimuli were presented using a battery-powered stereo-speaker (Logitech) through a USB DAC and audio amplifier (Fiio). The beginning of each trial was marked by a flashing white circle on the stimulus laptop screen which was detected by a photo-diode (Thorlabs, Inc.) attached to an auxiliary channel on the Intan recording system. The audio waveforms were recorded using a clip-on microphone (Movophoto, Inc.) which was connected to a pre-amplifier (Behringer) and digitized at 44,100 Hz by the Psychtoolbox module in MATLAB on the recording laptop. This microphone recording was also digitized on the Intan Technologies RHD recording system at 20,000 samples per second (along with the photodiode). The auditory stimulus, spoken non-word production response, and phoneme onsets were manually identified using the Audacity software package.

## Neural data analysis and decoding models

For preprocessing, the neural recordings from μECoG electrodes were decimated to 2,000 samples per second using an anti-aliasing filter followed by common-average-referencing (CAR). The electrodes with impedance greater than 1 MOhm or with recordings greater than 3 * log-RMS were excluded from CAR. Further, we removed trial epochs (after CAR) with excessive artifacts (greater than 10 RMS). All of our data analysis and neural decoding were performed using the following software: MATLAB R2021a, Python 3.7, Tensorflow 2.0, and Keras 2.4.

## Spectrogram visualization

We computed spectrograms for each electrode to determine the presence of speech-neural activity. The spectral information was extracted within a 500 ms analysis window (50 ms step-size) with 10 Hz frequency smoothing using a multi-taper time-frequency analysis method[8]. The resultant spectrograms were normalized with respect to a baseline period (500 ms window pre-stimulus across trials) by dividing the resultant spectrogram by its average baseline power-spectral-density to remove $1/f$ broadband activity. The normalized spectrograms were then averaged across trials and the log-power was computed around the speech utterance (−500 ms to 500 ms) to quantify neural activations in decibels.

$$HG - ESNR\,(dB) = 20 * \log_{10}\left(\sum_{t=-0.5}^{0.5}\sum_{f=70}^{150}\frac{Spect(t,f)}{\frac{1}{\tau}\sum_{\tau} Spect_{baseline}(\tau,f)}\right) \quad (2)$$

## Neural feature extraction

Neural high gamma signals (HG: 70 – 150 Hz) have been previously shown to characterize localized neural activity from cortical regions and are highly correlated with multi-unit firing[23,24]. They therefore serve as a robust marker for localizable neural activity. The pre-processed neural signals from each electrode were band-pass filtered into 8 center frequencies (logarithmically arranged) between 70 and 150 Hz[66]. The envelope of each frequency band was extracted using the Hilbert transform and was averaged across bands. The resultant

envelope was then downsampled to 200 Hz using an anti-aliasing filter and mean-normalized with respect to the baseline period (500 ms window pre-stimulus across trials). For each trial, we used a one second window of normalized HG centered at phonemic onset (−500 ms to 500 ms) to represent speech neural activity. The low frequency signals (LFS) were band-pass filtered between 1 and 30 Hz using a two-way least-squares finite-impulse-response filter, downsampled to 200 Hz, and mean-normalized to the baseline period.

## Cortical state space analysis using SVD-tSNE

We used a singular value decomposition (SVD) to transform normalized HG activity into a low-dimensional sub-space[41,67]. SVD was carried out on the covariance matrix of the HG activity to decompose highly covarying space-time activations across the array.

$$\mathbf{H} : \mathbf{U} \, X \, \mathbf{ET} \tag{3}$$

U – number of utterance trials, E – number of significant HG μECoG electrodes, T – time duration of HG.

$$E\left[\mathbf{H}^{\mathsf{T}}\mathbf{H}\right] = \mathbf{Q}\lambda\mathbf{Q}^{T} \tag{4}$$

The eigen-vectors (**Q**) served as basis for spatio-temporal activations with each trial receiving principal component (PC) scores corresponding to the eigen-vectors. To control for HG variability across subjects, we selected the set of ordinal eigen-vectors that cumulatively explained 80% of neural variance. The resultant PC scores were further projected to a two-dimensional subspace using t-distributed Stochastic Neighbor Embedding (t-SNE; Barnes-Hut algorithm, perplexity = 30)[45], a non-linear affine transformation technique for visualizing high-dimensional data. Finally, the trial embeddings were coded to represent either phoneme or articulatory features.

To evaluate the clustering of trials with respect to either articulatory features or phonemes, we performed a silhouette analysis (Euclidean distance)[68] for each state-space trial that measured the closeness of trial to a given cluster with respect to the other clusters. We further determined two metrics to evaluate the clustering strength of the cortical state-space: 1) silhouette ratio: ratio of the number of trials with positive silhouette score to the total number of trials, and 2) silhouette score: the average silhouette values of positive silhouette trials. Finally, to motivate the benefit of high-resolution neural recordings, we repeated the cortical state-space analysis on spatially sub-sampled electrodes extracted using Poisson disc sampling[32], by spatially constraining the minimum distance ($r = 2*\sqrt{\frac{(X*Y)}{n*3.5}}$) between subsampled electrodes (n).

## Linear decoding model using SVD-LDA

We constructed a linear-classification model with 20-fold cross validation to decode phonemes from normalized HG neural activations. First, we used SVD to compress the HG matrix (training set) to a low-dimensional subspace as explained in the previous section (Eq. 3 & 4). Then, we utilized the resultant principal component scores to classify phonemes using a supervised linear-discriminant (LDA) model. The optimal number of eigenvectors (or principal components) was identified using a nested cross-validation procedure, where a separate 20-fold cross-validation with grid-search was conducted on the training set. The nested cross-validation was conducted on every training fold. The overall decoding accuracy was calculated by calculating the proportion of correctly predicted labels in the test set. We trained decoding models to decode phonemes for each of the three positions in the non-word (P1, P2, and P3) and by collapsing phonemes across all positions.

To characterize decoding error, we examined values in the confusion matrix to test for confusion between phonemes that share similar phonological features. To characterize each phoneme, we created a 17-bit categorical vector based on work by Chomsky and Halle to represent phonological features[47]. We then computed the Hamming distance (bits) for each phoneme pair to determine the pairwise phonological distance. Phonemes with similar phonological features (e.g., /p/ vs. /b/) differed by 1-bit, whereas phonemes with dissimilar features differed by a maximum of 9 bits (e.g., /p/ vs. /i/). Finally, we mapped the percentage misclassification onto phoneme distance to evaluate phonological confusion of the neural signals and our classifier. The decoding error in terms of phonological distance was calculated using the equation:

$$e = \frac{1}{9}\sum_{i,j}\frac{1}{2}C_{i,j}D_{i,j} \tag{5}$$

e is phoneme error (bits), C is normalized confusion matrix, D is phonological distance (bits).

In addition, we used the decoding strategy to infer the spatial and sequential representation of phonological features. To infer the spatial encoding of features, we first grouped nine phonemes into four articulator features (low: /a/, /ae/, high: /i/, /u/, labials: /b/, /p/, /v/, and dorsal: /g/, /k/). We then trained an SVD-LDA decoder model (four-way) on each electrode to decode these articulatory features. The Area Under the Receiver Operating Characteristic Curve (ROC-AUC or AUC) was calculated for each electrode to quantify the specificity of articulatory feature encoding.

To examine the sequential representation of phonemes, we used temporal decoder models to train and test at every time-segment t, with respect to the speech utterance[69]. With this temporal modeling strategy, each SVD-LDA model was trained on sequential time-segments with respect to utterance onset, and the accuracy values were evaluated on the same time-segments. Through this strategy, we could assess the presence of temporal phoneme information present in the raw time signal. In both training and testing, the time-segments consisted of 200 ms windows (10 ms step size) ranging from −1000 ms to 1000 ms aligned to the utterance. The resultant time-series of accuracy values was used to determine the temporal activation of phoneme representations during speech production.

## RNN based seq2seq model for phoneme sequence decoding

We constructed a seq2seq encoder-decoder network to simultaneously decode all phonemes in a non-word from HG activations[21]. The neural-network contained three layers to translate HG activity to phoneme sequences: 1) Temporal convolutional layer, 2) Encoder recurrent neural network (RNN), and 3) Decoder RNN.

1. *Temporal convolutional layer*: The first layer contained a set of sequential kernels to transform HG into downsampled features. This convolutional layer to model temporal features was essential to reduce memory loading on the encoder RNN. We fixed both the kernel width and the hop length to 50 ms and no non-linearity was applied to the convolution kernels. The convolution layer resulted in linear temporal decomposition of HG.

2. *Encoder RNN*: The second part of the network contained a single layer of RNN units to model the output of convolutional layer at each time step. We used bi-directional LSTM (Long-Short-Term-Memory)[70] cells to represent each RNN unit, and the memory states of both forward (right arrows) and backward (left arrows) layers were averaged to return the final memory state as the output.

3. *Decoder RNN*: The final layer contained a sequence of a unidirectional RNN to predict one-hot encoded phoneme units at each position. The decoder RNN was instantiated by the final output state of the encoder RNN and was configured to return both the memory state and output sequences. The output values from each time-bin were then converted to phoneme probabilities through a dense neural net with SoftMax activation.

The architecture was implemented in Keras-Tensorflow[71]. The network was trained using the Adam[48] gradient descent optimization (learning rate = 1e-3; 800 epochs) by minimizing the categorical cross-entropy between true and predicted phoneme probabilities[48]. We included an L2 regularization term at both convolutional and recurrent layers to prevent overfitting. Overall, the architecture contained the following hyperparameters: 1) number of convolutional kernels, 2) number of RNN units, and 3) L2 penalty. For each subject, we held 20% of trials for testing and we performed training and hyperparameter optimization on the remaining 80% of trials.

During training (80% trials), the decoder RNN evaluated the phoneme probability at every position and the RNN input was set to one-hot encoded vector of phonemes at the previous step. For faster convergence and stable modeling, we utilized the "teacher – forcing" approach where we used the ground truth (true phonemes) as input to the decoder RNN[72].

We identified the optimal hyperparameters on the training-set (80% trials) for the neural-network using the keras-tuner platform[73]. For each subject, we randomly explored the hyperparameter space through 200 hyperparameter combinations. We then calculated the validation accuracy (10-fold cross-validation, three repetitions) for each combination, and identified the hyperparameter set with the maximum-validation accuracy. The hyperparameters for each subject are summarized in Supplementary Table 5.

During testing (20% trials), we used the trained decoder model (using optimal hyperparameters) to generate the most probable phoneme sequence by identifying the most likely phoneme at each position and used it as the input to the next position.

Overall, we evaluated the model by calculating its phoneme decoding accuracy between the tested and predicted phonemes. To validate the seq2seq model against the linear approach, we trained an SVD-LDA model (5-fold cross-validation; analogous to 20% held-out) and calculated the accuracy and phoneme error in decoding phonemes across all positions. Further, to quantify the influence of spatial resolution, we evaluated the decoding performance at different spatial resolutions through subsampling and training the encoder-decoder models on subsampled versions of the μECoG recordings.

### Objective assessment of acoustic contamination

We objectively quantified the presence of acoustic contamination in μECoG recordings by comparing spectral information between microphone audio channel and voltage time series of each electrode (convolution using a 200 ms Hamming window, followed by estimation of the power spectral density)[74]. We measured the contamination in each electrode by computing the correlation between audio and neural signals across all possible pairs of frequency bins, resulting in the audio-neural contamination matrix per electrode. The statistical significance of the correlation values in the HG band is estimated by comparing the mean diagonal of the contamination matrix (between 70 and 150 Hz) against the distribution of shuffled versions of the contamination matrix (10 000 iterations) and the resultant p-value was corrected for false discovery rate.

### Statistical analysis

We used multiple statistical methods to validate μECoG recordings for speech decoding. To identify electrodes with significant speech-neural activation, we compared HG power between the response and the baseline period, using a 10,000-iteration one-sided permutation test (corrected for false discovery rate). For the control subjects, we performed similar non-parametric test on SEEG/ECoG electrodes anatomically localized to SMC to identify significant speech-neural activations. To characterize the morphology of HG responses, we extended the above non-parametric test to estimate the significance at every time point during utterance. The resultant time-series

statistical scores were subjected to a second-level non-parametric temporal cluster[68]. For each electrode with significant clusters, we estimated the onset and durations of the cluster with respect to utterance. For ESNR comparisons to standard IEEG, we used one-sided Mann-Whitney U test to compare the HG power (during the speech production response) between the population of μECoG electrodes and the control IEEG. Finally, we used one-way ANOVA to examine the effect of spatial subsampling, as well as post hoc *t-tests* to identify subsampling groups with significant differences in means.

To quantify signal sharing between μECoG electrodes, a cross-channel Pearson correlation coefficient ($\rho$) was calculated between HG time-series (during the speech production response) and was linked to corresponding Euclidean distance ($x$). The mean correlation coefficient was identified at each unique spatial distance to assess the spatial relationship of HG between electrode pairs.

For phoneme decoding, the prediction accuracies were tested against chance using a binomial cumulative distribution test[46]. All population data are indicated with mean and standard errors, and we estimated the significance of model fits using linear regression.

### Reporting summary

Further information on research design is available in the Nature Portfolio Reporting Summary linked to this article.

## Data availability

The data from μECoG recordings that support the decoding analysis are available via the Data Archive for The Brain Initiative (DABI; accession code: https://dabi.loni.usc.edu/dsi/7RKNEJSWZXFW) under restricted access (restrictions will be exempted after appropriate modifications of the IRB protocol: Pro0072892). The access can be obtained upon written requests from the corresponding authors (gregory.cogan@duke.edu, j.viventi@duke.edu). Source data are provided with this paper.

## Code availability

The MATLAB files to perform decoding analysis are available through Zenodo at Suseendrakumar Duraivel. coganlab/micro_ecog_phoneme_decoding: v1.0. (2023) https://doi.org/10.5281/zenodo.8384194.

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

## Acknowledgements

J.V., G.B.C., S.P.L., D.G.S., A.H.F., S.R.S., S.D. were supported by NIH R01DC019498 and DoD W81XWH-21-0538, J.V., G.B.C., C.W., D.G.S., S.R.S., S.D. were supported by NIH CTSA UL1TR002553, J.V. and G.B.C. were supported by NIH UG3NS120172, G.B.C, J.V., S.P.L, D.G.S, and A.H.F were supported by NIH R01NS129703 and G.B.C., J.V., D.G.S, S.P.L., S.R.S., S.D. were supported by an internal Duke Institute for Brain Sciences Incubator Award. D.G.S. was supported by the NINDS Neurosurgery Career Development Program and the Klingenstein-Simons Foundation. We would like to thank Anna Thirakul for help with consenting and IRB compliance, Palee Abeyta for help with data collection for the in-unit patients with epilepsy, Seth Foster for helping with visualization software, and James Carter for help with intraoperative data collection. We would also like to thank the patients who graciously volunteered their time for this study.

## Author contributions

S.D. performed the neural feature analysis, developed speech decoding models, performed statistical assessments, and generated figures. G.B.C., and S.R. designed the experimental task. S.R., S.C.H., S.P.L., A.H.F., and D.G.S., were involved in surgical planning and performed intra-operative electrode implantation and neural data acquisition. C.H.C., C.W., K.B., and J.V. were involved in μECoG electrode design and setting up intra-operative data acquisition process. D.G.S., and S.R.S assisted in extra-operative data acquisition from SEEG and ECoG electrodes. M.T., S.R., C.H.C., J.V., and G.B.C contributed to neural feature analysis and the development of speech decoding models. S.D., J.V., and G.B.C., wrote the manuscript with feedback from the other co-authors. G.B.C. was involved in clinical supervision. J.V., and G.B.C. designed and supervised the human intra-operative study.

## Competing interests

The authors declare no competing interests.
