## [Peer Review File · Nature Communications]

High-resolution neural recordings improve the accuracy of speech decodingREVIEWER COMMENTS

Reviewer #1 (Remarks to the Author):

In this manuscript titled "High-resolution neural recordings improve the accuracy of speech decoding", the authors use micro-ECoG recordings in 4 able-bodied individuals to decode speech. The long-term goal of this research is to aid the development of brain-computer interfaces (BCIs) for individuals with a locked-in syndrome. The authors demonstrate that micro-ECoG offers superior spatial resolution and higher signal-to-noise ratio and therefore provides more accurate phoneme decoding and encoding of articulatory speech features compared to clinical (10 mm pitch) and high-density (3-4 mm pitch) ECoG grids.

Overall the presented work is well described and extensive analyses were performed to display multiple measures of decodability. Yet, several concerns dampen my enthusiasm for the work, and they have not been addressed in the manuscript. For one, results are taken to provide firm evidence that the micro ECoG recordings bring us much closer to decoding speech in BCI use-cases. This is not obvious from the data in that decoding is not that high for 9 phonemes, especially the ones embedded within (nonsense) words. In BCI use cases, phonemes are typically embedded (compare to P2 and P3) and are more numerous than 9. As authors state, decoding becomes less when motor programs of phonemes are more alike (eg p and b). Second, in abled participants the somatosensory cortex is activated by sensory feedback, which is absent in non-abled persons. Previous studies have shown that attempted speech is much more difficult to decode (eg Ed Changs group) which likely further reduces decoding performance. Third, many analyses are performed on rather limited data, some of which essentially measure the same phenomenon (eg subsampling), somewhat inflating the impact of the findings. In general, the benefit of ultra-high density configurations is clearly proven in this work (and getting reasonable results from a DNN on such little data does show the robustness of the micro-ECoG signal), but that was to be expected given better decoding with 4 mm pitch grids compared to 10 mm pitch and other publications on the detail of sensorimotor representations (mentioned in the introduction). Hence the strength of this paper is rather incremental than a leap forward. In all, I think the results are somewhat over-interpreted. I list major and minor comments below.

MAJOR

- 1) Section 1: The authors focus entirely on high gamma, although a lot of the latest literature (Anumanchipalli et al. 2019, Nat; Sun et al. 2020, J Neural Eng; Proix et al. 2022 Nat Comm) points towards the importance of low frequencies as well. Could the authors provide a reasoning for leaving low frequencies out of the analysis or add this to the limitations of their study?
- 2) Section 2: The manuscript does not seem to discuss the anatomical locations of electrodes with highest high gamma modulation, largest contribution to decoding and best encoding of articulatory features. The consistency of anatomical location of such electrodes across subjects and its relation to the homunculus or the literature is also not explicitly discussed. Are the most informative electrodes on the dorsal or ventral part of the sensorimotor cortex, on the motor or somatosensory part?
- 3) Section 2: The authors do not seem to have performed analyses confirming lack of acoustic contamination in their neural signals (see Roussel et al. 2020, J Neural Eng). Given that the data are acquired in able-bodied individuals, it would strengthen the results if the authors could provide this check.
- 4) Section 2: What can the authors say about the amount of time necessary to decode each phoneme? Was there a fixed temporal windows around each phoneme? What time window led to the best performance? Did it differ across positions? It appeared that the authors did not explicitly test that.
- 5) The authors seem to make no distinction between motor articulatory features and phoneme language features. This may be because, looking at Supplementary Table 6, the utterance classification would not have been as high as the already reported work with complete words because they cannot use a language model to improve them. The authors argue that this is future

work, but the idea of turning ok phoneme classification into good word classification is not novel. Could the authors comment on their focus on phoneme features compared to true motor articulation features, such as position and trajectories of speech articulators?

6) In Section 1 it is mentioned that the grids are placed over SMC, but electrodes included for analyses are not restricted to SMC. For subject S4 with a 256-electrode grid this would cut the number of electrodes by more than half. The authors argue that the fact that S4 does better than S1 and S2 when an equal amount of data is used (S4 had only one run), implies that more electrodes help. In fact, S4 does not have more electrodes over SMC and the extra electrode over Broca do seem to contribute to the classification results, confounding the comparison.

7) Section 2.1: How accurately are grids coregistered with individual anatomy for the 3 128-grids slid through the burr hole? It seems unlikely that the 200 Micron electrodes could be distinguished on CT (no images shown). Why not use one of the localization techniques available (eg Gridlock, Branco et al, Neuroimage 2018). Supplementary Fig 2 shows equal HG power in electrodes over sulci which deviates from papers with (more) accurate co-registration (eg the Branco paper)

8) Section 2.2: Better decoding of P1 is most likely due to phoneme transitions in P2 and P3. In BCI applications most phonemes will be in transitional positions, especially when people do not pause between words. This limits the decoding performance 'in real life' and should be discussed.

9) Section 2.2 and Fig 4C: tasks appear to differ between experiments given only 4 phonemes were included. Is it possible that phonemes of the ECoG and sEEG experiments were embedded and included phoneme transitions? (Hence the more accurate comparison should be average of P2 and P3)

10) Idem: the comparison in performance to clinical ECoG and sEEG is weak because in those subjects the SMC coverage is likely to be very sparse (especially sEEG) so the lesser performance is not very meaningful in that if you were to implant clinical scale grids for BCI the coverage would be much better.

11) Section 2.2 and Fig 5A: subsampling does not take into account that multiple electrodes are positioned over sulci. These do not contribute much to decoding including all electrodes but they can disproportionately contribute their noise to randomly subsampled configurations

12) Supp Fig 7C: The graph seems to plateau after 3-4 classes. I would rephrase the claim that with number of classes even more than 9 micro-ECoG is likely to provide larger decoding benefits compared to less dense ECoG grids.

MINOR

13) In Section 2.1 please explain the task better in the results section (info given in support table 2) and number of repetitions of nonsense words and phonemes

14) Line 105 and Fig 1: grid dimensions do not agree: fig 1: 12x22, line 105 12x24, both '256 channels;'

15) Line 126: the authors state that the increase in high gamma aligned to speech onset (-250 ms to 250 ms around speech onset), however Figure 2B (and possibly 2A) seems to show increases from -1 second prior to speech onset. Can the authors clarify this?

16) Line 142: Is the correlation calculated on baseline data or task data? This matters in that the latter should yield high correlations by default

17) Line 184: Could the authors specify how many principal components (that altogether explain 80% variance) were used for decoding ?

18) Line 186: Please explain what 'position' means in line 186

19) Line 332: Can the authors elaborate on why they used a bidirectional RNN over a unidirectional one? For a real-time system with a BCI user, a unidirectional RNN may be preferred. Besides, what would be the rationale for expecting the information about the present phoneme to be in neural signals almost a second later?

20) Line 431: It looks like the authors did not use any statistical information about phoneme sequences in their RNN. Please make this more explicit in the results section as well.

21) Fig 2A,B: are data from averaged trials?

22) Supp Table 4: What coordinate system is used for X,Y coordinates ?

Reviewer #2 (Remarks to the Author):

The authors investigate how speech decoding can be improved with μ ECoG electrodes that have a higher spatial resolution compared to standard and high-density ECoG grids. In the present study, the authors conducted speech production tasks with four patients who were implanted with μ ECoG grids to identify phonemes in the neural data via linear (LDA) and non-linear (Neural Network) classification techniques. Decoding results are compared to those acquired from ECoG grids with 4mm and 10mm inter-electrode spacing, as well as to stereotactic EEG. The authors give a comprehensive analysis in which aspects μ ECoG grids might provide superior information for speech decoding.

This is a strong paper, and the results show significant advances to the field. Overall, the manuscript is well-written, experimental details are clearly presented, the data analysis is properly documented, and all figures are well embedded into the main text and further underline their claims. The work presented in this manuscript is original and meets the expected standards in the field. I only have one moderate and a few minor comments / questions before the manuscript is ready to be published.

In section 2.1 the authors describe the subject pool and which tasks they conducted. In line 116, it is not clear to what part in the experiment design the term "utterance" refers to. Please clarify with respect to the experiment tasks how (average) utterance duration differs from repeating the auditorily presented words. Maybe the term can also be visualized in Figure 1E.

In section 2.1 the authors investigate how articulatory features are organized in the neural space. I suggest adding one sentence to the main text describing how the articulatory features were inferred. This information is a bit hidden in the Methods section, and in section 2.2 about the phonological distances.

In Figure 2A, the authors visualize spectral activity from each electrode for subjects S1 and S4. Based on the main text, it is not clear to me what exactly is represented in the spectrograms: Is it the average activity of a particular nonword token, or representing HG of speech activity in general? I suggest adding one sentence clarifying that.

In Figure 2C, the authors mention dotted lines. However, in the figure there are no dotted lines, and I believe they refer to the solid ones.

In section 2.1, the authors use poisson disc sampling to examine the contributions of the higher spatial resolution by repeating the state-space analysis on a subsampled spatial resolution. I am not fully convinced if the poisson disc sampling method is appropriate here as it might result in electrode clusters which are spatially aligned in a closer distance to what is capable of standard or high-density ECoG (as indicated by Fig. 4S). To my understanding, the r-value (radius) from the poisson disc sampling method should be chosen to be either 4mm or 10mm to provide comparable results to currently used grids, but this would also enable much lower spatial resolutions as poisson disc sampling uses the spherical annulus between r and $2r$ for selecting the next electrode,

and that could bias the analysis in favor of the proposed method. Happy to stand corrected on this ground.

In section 2.2, please clarify in the main text (one sentence should be sufficient) what you mean with all positions, position one, position two and position three to avoid ambiguity. It is not clear from the main text that it refers to the position within the nonword of the CVC or VCV.

Language: The manuscript uses the term μ ECoG extensively in the first part, but switches to high-density μ ECoG for the second (and then back). I suggest keeping it consistent. I also detected a few minor typos: ... "my" median decoding accuracy in the caption of Figure 5S, "um" diameter in caption of Figure 1.

Reviewer #3 (Remarks to the Author):

In "High-resolution neural recordings improve the accuracy of speech decoding", Duraivel et al. compared μ ECoG recordings with ultra-high density spatial resolution in four participants to standard ECoG/sEEG recordings to determine whether and how decoding speech information might benefit from increased spatial resolution afforded by μ ECoG. Overall, I think the authors have convincingly shown the benefits of using higher resolution arrays for speech decoding and for resolving spatiotemporal neural activity in general. I also appreciate the analyses comparing both downsampled or spatially smoothed versions of the μ ECoG array data to typical sEEG and ECoG, as this provides additional context for why higher density ECoG may result in better information for both encoding and decoding analyses. I have a few major and minor comments that I believe will improve the manuscript in its current form.

Major Comments:

1. The authors used a nonword task for their repetition task in the patients, but it is not explained why this was chosen over a word repetition or other closed vocabulary task, as is used for typical speech decoding experiments. Some additional text regarding the rationale behind this choice would be helpful.
2. Information about the macro ECoG/sEEG recordings and the placement of arrays is needed. At the moment, only a text description saying that the electrodes were placed in SMC or related areas is mentioned, but it is not possible to directly compare the placement of these electrodes to the placement of the μ ECoG grid. With μ ECoG grids, if the overall array size is smaller, it will be necessary to choose a more narrowly defined anatomical region from which to record. As seen in Figure 2 and Supplementary Figure 2, the arrays cover different anatomical regions of interest, with many of the strongest responses arising from postcentral and precentral gyrus. For sEEG, it is unclear how similar responses might look and whether they are sampling from the same region. Although I appreciate that these arrays should have higher resolution than most sEEG of this area, having a more direct comparison of the anatomy would be helpful.
3. In Figure 3 and Supplementary Figure 3, the authors show the spatiotemporal high gamma across the arrays in all four subjects with μ ECoG, and find a striking difference in patterns. How does this pattern compare to known topography of jaw, larynx, tongue, etc. as shown by, e.g. Bouchard et al. Nature 2013? While this paper and others are cited, it is not explained whether their current results show similar somatotopy or not.

Minor Comments:

1. In Figure 3, the authors show the difference between spatiotemporal activity for four different phonemes, which is further expanded in Supplementary Figure 3. It would be helpful for the authors to point out some of the more subtle differences that can be observed between phonemes that differ by only one feature, for example, /b/ vs. /p/, which differ by voicing, and there does seem to be a spatial pattern of activity in /b/ that overlaps somewhat with voiced compared to voiceless phonemes.
2. For Supplementary Figure 2, I appreciate seeing the diversity of spectrotemporal information

afforded by using μ ECoG grids – the differences are quite striking. However, I do wonder why the MNI brain was chosen for viewing, since there are only four participants, and it would be more accurate to show the arrays on the native brain.

3. For Supplementary Figure 3, why is there a * and # after S3 and S4?

4. Please consider enlarging font sizes for the smallest fonts for Figures 1-6. Especially, for example, the legend in Fig. 6C and the axis labels in 6D are difficult to read at the current size.

Reviewer #1 (Remarks to the Author):

In this manuscript titled “High-resolution neural recordings improve the accuracy of speech decoding”, the authors use micro-ECoG recordings in 4 able-bodied individuals to decode speech. The long-term goal of this research is to aid the development of brain-computer interfaces (BCIs) for individuals with a locked-in syndrome. The authors demonstrate that micro-ECoG offers superior spatial resolution and higher signal-to-noise ratio and therefore provides more accurate phoneme decoding and encoding of articulatory speech features compared to clinical (10 mm pitch) and high-density (3-4 mm pitch) ECoG grids.

Overall the presented work is well described and extensive analyses were performed to display multiple measures of decodability.

We appreciate the reviewer’s support of our efforts in employing high-density μ ECoG grids to decode speech.

Yet, several concerns dampen my enthusiasm for the work, and they have not been addressed in the manuscript. For one, results are taken to provide firm evidence that the micro ECoG recordings bring us much closer to decoding speech in BCI use-cases. This is not obvious from the data in that decoding is not that high for 9 phonemes, especially the ones embedded within (nonsense) words. In BCI use cases, phonemes are typically embedded (compare to P2 and P3) and are more numerous than 9. As authors state, decoding becomes less when motor programs of phonemes are more alike (eg p and b).

We recognize the reviewer’s concerns in our present work. Our overall goal was to demonstrate the utility and fully characterize the performance benefits of using μ ECoG to decode speech for future BCI work. In our study, we demonstrated the first case of using high-density cortical electrodes (less than 1.5 mm electrode spacing) that achieved accurate phoneme decoding with cortical recordings in the intra-operative settings and with overall experiment times less than 15 minutes (spoken duration at 2 minutes). Even with this time-constrained experimental procedure, our decoding accuracy is maximum for 9 phonemes and this maximal performance is dependent on spatial sampling, spatial coverage, contact size, decoding time-window, and training duration, all of which are essential factors in designing BCI for speech prosthetic applications. We acknowledge the reviewer’s concern about our decoding score for 9 phonemes, and especially with phonemes embedded within non-word stimuli. However, with existing studies¹⁻⁶ varying in trial numbers (spoken duration) and number of phonemes to decode, we couldn’t directly benchmark our decoding results. Therefore, we sought to in-house comparison by simulating different versions of intracortical surface recording (in section 2.2) and observed 17% to 29% improvement from 4-mm to 10-mm electrode array configurations, respectively. Also, we characterized the decoding error^{2,4,5} (in section 2.2) and observed that the phoneme misclassifications are systematic with respect to speech articulation and not a characteristic of our recording ability. We quantified this performance reduction by calculating decoding error in phonological space as it relates to speech articulatory structure. This systemic phoneme error, to the reviewer’s point, explains less decoding for more-alike phonemes. In addition, this error calculation could enable future BCIs to predict phoneme confusions on an “a-posteriori” basis and perform post-hoc error correction⁷. Further, we investigated causes of this reduction in performance for embedded phonemes under section 2.2 and section 2.3 and identified that the decoding performance can be impaired by neural contamination with co-articulatory effects and discrepancies in accurately marking phoneme onsets (in keeping with Wilson *et al.*, 2018⁴ and Mugler *et al.*, 2014²). We found that this variability in manually marking phoneme onset timing reflects an important factor for BCI use cases and so we therefore developed a speech decoding framework (in section 2.4) that does not require explicit markings of embedded phonemes within non-words. This model led to improved decoding performance.

Second, in abled participants the somatosensory cortex is activated by sensory feedback, which is absent in non-abled persons. Previous studies have shown that attempted speech is much more difficult to decode (eg Ed Changs group) which likely further reduces decoding performance.

We recognize the reviewer’s concern about neural contributions that are directly resulted from sensory feedback. We agree with the reviewer that the sensory feedback will be absent in non-abled persons, which would make the decoding of attempted speech more difficult. To address this concern, we added a new supplementary fig. 5 to characterize the temporal morphology of high-gamma (HG) responses from each μ ECoG electrode. We identified time-points with significant

activation as compared to baseline ($p < 0.05$, cluster correction based one-sided permutation test). From each significant cluster, we determined both the onsets and duration of HG activations with respect to the speech utterance onset. Across all subjects we show that 77.4% of electrodes (336/434) with HG activations have significant clusters that start before the utterance onset, indicating that our μ ECoG electrodes measured earlier motoric activations leading to speech (that are not necessarily feedback- Supplementary Fig. 5).

Further, studies examining functional characteristics of the sensorimotor cortex in tetraplegic patients have revealed that cortical representations of motor commands are intact even years after the loss of motor functions^{8,9}. Moses *et al.* (2021) were able to decode attempted speech from an anarthric patient using ECoG with 4-mm electrode spacing indicating that speech cortical functions were still preserved even after 15 years of anarthria¹⁰. In our present results on speech decoding, we predict that our high-resolution μ ECoG can better resolve preserved motor commands as compared to conventional ECoG, which in turn can improve the performance of attempted speech decoding. Future studies will involve implanting our high-resolution μ ECoG in patients with debilitating motor disorders who have lost/reduced ability to speak to explicitly quantify the performance of attempted speech decoding.

We made the following changes to the manuscript:

We included Supplementary Fig. 5, to quantify the timing and temporal morphological features of HG activations aligned to the speech utterance.

We included the following text in Results under section 2.1.

These electrodes exhibited spatially varying characteristics of HG activations with 77.4% of these electrodes (S1 107/111, S2 107/111 S3 34/63, and S4 88/149) were active before the utterance start, indicating that μ ECoG electrodes measured earlier motoric activations leading to speech (Supplementary Fig. 5).

Third, many analyses are performed on rather limited data, some of which essentially measure the same phenomenon (eg subsampling), somewhat inflating the impact of the findings.

We recognize the reviewer's concern with measuring and quantifying similar neural populations at subsampled versions. However, we argue that our μ ECoG electrodes record unique phoneme-related neural activation that was robust across speech trials. These micro-scale speech neural activations required finer than 2-mm spatial sampling to achieve the maximum decoding accuracy for the 9-way phoneme decoder. To the reviewer's point, measuring similar neural phenomena at higher spatial resolutions could have saturated the decoding performance. To test this hypothesis, we quantified the decoding performance of all possible N-way decoders (N = number of phonemes) and measured the sampling resolution necessary to reach the desired decoding (Supplementary Fig. 14C). Higher spatial resolutions were required to achieve maximum decoding with more unique phonemes, indicating that phoneme related neural populations are encoded at higher spatial resolutions.

In general, the benefit of ultra-high density configurations is clearly proven in this work (and getting reasonable results from a DNN on such little data does show the robustness of the micro-ECoG signal), but that was to be expected given better decoding with 4 mm pitch grids compared to 10 mm pitch and other publications on the detail of sensorimotor representations (mentioned in the introduction). Hence the strength of this paper is rather incremental than a leap forward. In all, I think the results are somewhat over-interpreted. I list major and minor comments below.

We would like to stress the importance of micro-scale recording feature of our high-density cortical electrodes compared with standard clinical ECoG grids and penetrating microelectrodes. To the reviewer's point, while it is likely to expect decoding performance increases as compared to previous studies, we emphasize that the focus of our current work is to validate our thin-film μ ECoG electrode arrays for neural speech decoding. These arrays possess the ability to record fine-scale neural activations (undetectable with standard clinical grids), are flexible and movable enough to target desired cortical regions and can be scaled to wider coverage (unlikely with penetrating microelectrodes), all of which are essential factors for designing BCI systems. In this paper, we exploited these features to improve speech decoding and established the relationship between decoding performance and high-dimensional recording capabilities. We believe that our findings place high-density μ ECoG as a competitive neural recording technology for speech BCI systems. In the future, we will demonstrate

real-time BCI decoding using high-density, high channel count μ ECoG in patients with debilitating motor disorders who have lost/reduced ability to speak.

We thank the reviewer for their detailed review and feedback on our manuscript. Responses to comments and revisions are addressed point-by-point, below.

MAJOR

1) Section 1: The authors focus entirely on high gamma, although a lot of the latest literature (Anumanchipalli et al. 2019, Nat; Sun et al. 2020, J Neural Eng; Proix et al. 2022 Nat Comm) points towards the importance of low frequencies as well. Could the authors provide a reasoning for leaving low frequencies out of the analysis or add this to the limitations of their study?

We re-applied our decoding analysis using low frequencies signals¹¹ (LFS: 1 - 30 Hz) and using a combination of LFS and HG power. As shown in the Supplementary Fig. 9, we observed that LFS enabled significant, yet lower decoding scores compared to decoding scores from HG power, in all subjects. The combination of LFS and HG power resulted in decoding performance slightly lower than HG power alone (except S3). This analysis indicates that LFS encoded similar and not additive phonemic information compared HG power in our speech production task.

Studies that examined cortical oscillations associated with LFS, have argued that they are important for the tracking of the temporal structure of speech, specifically for prosodic and syllabic rhythms during speech processing¹²⁻¹⁵. Consequently, the reduced performance in our study may be attributed to our stimuli design of single non-word speech tokens as opposed to naturalistic stimuli with varying syllabic/prosodic information. We predict that the decoding contribution from μ ECoG LFS would be higher for more naturalistic stimuli. Our future work will include designing speech tasks under more naturalistic conditions for real-time BCI applications.

We attached our results as a Supplementary Fig. 9 and included the following changes to the Methods, Results, and Discussion.

Methods (Section 4.6.2)

The low frequency signals (LFS) were band-pass filtered between 1 and 30 Hz using a two-way least-squares finite-impulse-response filter, downsampled to 200 Hz, and mean-normalized to the baseline period.

Results (Section 2.2)

This decoding performance was specific to HG information alone as adding low frequency signals (LFS) did not result in performance increases (Supplementary Fig. 9).

Discussion

The decoding scores were obtained from the HG band (70 to 150 Hz) and adding LFS as additional features did not improve the performance (Supplementary Fig. 9). LFS were associated with the temporal structure of speech (prosodics and syllabic information)¹²⁻¹⁵, and the lack of performance improvements may be attributed to the stimuli design of single non-word speech tokens, compared to naturalistic stimuli with varying syllabic/prosodic information.

Section 2: The manuscript does not seem to discuss the anatomical locations of electrodes with highest high gamma modulation, largest contribution to decoding and best encoding of articulatory features. The consistency of anatomical location of such electrodes across subjects and its relation to the homunculus or the literature is also not explicitly discussed. Are the most informative electrodes on the dorsal or ventral part of the sensorimotor cortex, on the motor or somatosensory part?

We have now mapped the electrodes onto subject specific brains using each subject's preoperative MRI and intraoperative CT imaging (or Brainlab coordinates for S4). We identified the pre-central, central, and post-central sulcus for each subject (Supplementary Fig. 2). We added supplementary figures 5 and 6 in which we show the spatial correlates of HG activations for each subject. We also overlaid electrode array for each subject onto their pre-central, central, and post-central sulcus (where applicable - Supplementary Figs. 4, 5, 6, 11, and 15) and added the following text to the Discussion section.

Discussion

Even with the time-constrained experimental procedure (up to 15 minutes of recording duration/ 1.04 minutes of total spoken duration; **Fig. 1E**, Supplementary Table 3), our μ ECoG electrodes recorded neural signals in SMC that produced significantly higher HG-ESNR as compared to standard intracranial methods (ECoG/SEEG). Electrodes with HG activations spanned both the pre-central (motor) and post-central (sensory) gyrus across the SMC for S1 (Supplementary Fig. 2 & 4). Subject S2 and S3 had predominant coverage in the pre-central and post-central gyrus, respectively. Subject S4 had a 256-channel array that extended anteriorly from SMC to the inferior frontal gyrus, however, most HG activations were confined to SMC.

On a subject-specific scale, we observed high-resolution speech articulator maps, similar to previously established group-level organization¹⁶. We observed that the surface area for each articulator map ranged from 11 mm² to 80 mm² depending on the location and coverage of each μ ECoG grid (Supplementary Table 4). Overlaying these spatial clusters on each subject's individual brain, we made the following anatomical observations: S1 and S3 exhibited labial clusters near the posterior region of post-central (sensory) gyrus, and dorsal-tongue clusters (only S1) near the anterior-inferior region. S2 displayed labial and dorsal-tongue clusters over the pre-central (motor) gyrus; and S4 demonstrated distinct articulatory clusters within pre-central gyrus, indicating that electrodes with high speech articulator information were contained within SMC. This anatomical clustering is similar to previously identified somatotopic representations, that revealed a posterior clustering of lip articulators and an anterior clustering of tongue articulators with respect to central sulcus¹⁶.

3) Section 2: The authors do not seem to have performed analyses confirming lack of acoustic contamination in their neural signals (see Roussel et al. 2020, J Neural Eng). Given that the data are acquired in able-bodied individuals, it would strengthen the results if the authors could provide this check.

As suggested by the reviewer, we performed the analysis outlined in Roussel *et al.* 2020¹⁷ and objectively quantified the presence of microphone signals by calculating the correlation between neural signals and microphone signals. While we observed some contamination of microphone signals in our first subject (S1), the contamination was significant only in higher frequency regions (greater than 200 Hz). We did not observe substantial contamination in other subjects. Since in our study we were interested in characterizing neural information within the HG band, we objectively quantified the presence of contamination in this band using a permutation shuffle test and observed no significant contamination in any electrodes across all subjects.

We added the results of our contamination analysis in Supplementary Fig. 3 and included following sections in Results and Methods:

Results (Section 2.1)

To confirm the absence of acoustic contamination in the neural data, we objectively examined the recordings for microphone contamination and did not observe a significant presence of microphone signals in our neural band of interest, across all subjects (except S1 at higher frequencies greater than 200 Hz; Supplementary Fig. 3).

Methods (Section 4.6.6)

We objectively quantified the presence of acoustic contamination in μ ECoG recordings by comparing spectral information between microphone audio channel and voltage time series of each electrode (convolution using a 200 ms Hamming window, followed by estimation of the power spectral density)¹⁷. We measured the contamination in each electrode by computing the correlation between audio and neural signals across all possible pairs of frequency bins, resulting in the audio-neural contamination matrix per electrode. The statistical significance of the correlation values in the HG band is estimated by comparing the mean diagonal of the contamination matrix (between 70 and 150 Hz) against the distribution of shuffled versions of the contamination matrix (10 000 iterations) and the resultant p-value was corrected for false discovery rate.

4) Section 2: What can the authors say about the amount of time necessary to decode each phoneme? Was there a fixed temporal windows around each phoneme? What time window led to the best performance? Did it differ across positions? It appeared that the authors did not explicitly test that.

We performed the analysis as suggested by the reviewer and calculated the decoding scores at increasing time-window size for all subjects and phoneme positions. We added Supplementary Fig. 12 to demonstrate the relationship between decoding performance and time-window size and included the following changes to Results and Discussion:

Results (Section 2.2)

The decoding results used a fixed time-window (-500 ms to 500 ms) across phoneme onsets. To determine the optimal time-window required to decode phonemes, we calculated the decoding scores at varying time-windows for all subjects and phoneme positions (Supplementary Fig. 12). The decoding performance increased with longer windows until saturation at around 500 ms. This saturation point was specific for each subject and phoneme position, and the optimal decoding values were greater than or equal to values observed with fixed time-windows. S3 and S4 exhibited a reduction in decoding beyond the optimal point, indicating that performance could be impaired by adjacent phonemes at low SNR and reduced data duration.

Discussion

The decoding performance decreased for phonemes embedded within the non-word, most likely due to discrepancies from marking onset times of embedded phonemes and co-articulatory effects that arise during phoneme transitions. To investigate these effects, we examined the amount of time necessary to decode each phoneme (Supplementary Fig. 12). We observed increased decoding performance with larger windows until saturation at around 500 ms similar to previous results^{2,4}. This optimal window was specific for each position indicating a need for decoding models that do not require explicit phoneme onsets. This decoding window selection, however, is essential for online systems with an optimal tradeoff between speed and accuracy¹⁸.

5) The authors seem to make no distinction between motor articulatory features and phoneme language features. This may be because, looking at Supplementary Table 6, the utterance classification would not have been as high as the already reported work with complete words because they cannot use a language model to improve them. The authors argue that this is future work, but the idea of turning ok phoneme classification into good word classification is not novel. Could the authors comment on their focus on phoneme features compared to true motor articulation features, such as position and trajectories of speech articulators?

In this study, we are interested in motivating the importance of high-resolution μ ECoG for speech decoding. We therefore quantified our ability to decode both phonemes (fundamental unit of speech) and motor articulatory features using non-words to limit any effects of higher-order language processing. We felt that this demonstration would more accurately capture and reflect the benefits of using μ ECoG recording devices. Our future work will build upon this result with the aid of large language models (LLMs).

We included the following text in the Discussion section to motivate the usage of phonemes and articulatory features:

Discussion

We performed speech decoding by classifying phonemes from all positions within the spoken non-word. Using non-word stimuli in the speech task enabled us to remove the effects of higher-order language processes and to decode phonemes and articulatory features to clearly assess the benefits of using high-density μ ECoG to decode speech features from neural recordings.

6) In Section 1 it is mentioned that the grids are placed over SMC, but electrodes included for analyses are not restricted to SMC. For subject S4 with a 256-electrode grid this would cut the number of electrodes by more than half. The authors argue that the fact that S4 does better than S1 and S2 when an equal amount of data is used (S4 had only one run), implies that more electrodes help. In fact, S4 does not have more electrodes over SMC and the extra electrode over Broca do seem to contribute to the classification results, confounding the comparison.

To mark the distinction between SMC and more anterior regions in subject S4, we overlaid the location of the electrode array onto the pre-central sulcus (shaded red curve – Supplementary Fig. 11). Electrodes with significant HG activation (yellow), were confined to the pre-central gyrus. Further, when examining the speech articulator maps (Supplementary Fig. 15), we observed that all the speech articulatory map clusters were observed posterior to the pre-central sulcus, indicating that high information speech articulatory electrodes were restricted to SMC.

To the reviewer's point, we observed 16 electrodes with significant but weaker SNR, anterior to the pre-central sulcus. We directly compared the decoding score from these electrodes and obtained non-significant decoding from these channels (Supplementary Fig. 11), indicating that the strong classification performance is from electrodes over SMC.

We made the following changes to address the reviewer comments.

We modified the electrode array map of S4 in all supplementary figures by including the red curve to indicate the location of the pre-central sulcus.

We added the Supplementary Fig. 11 to compare the decoding performance of electrodes from SMC and inferior frontal gyrus.

We made the following changes to the Results and Discussion.

Results (Section 2.2)

Although higher spatial coverage of S4 enabled recordings from both inferior frontal gyrus (Broca's area) and SMC, significant decoding was largely specific to electrodes over SMC (Supplementary Fig. 11).

Discussion

Subject S4 had a 256-channel array that extended anteriorly from SMC to the inferior frontal gyrus, however, most HG activations were confined to SMC.

Further, S4's decoding performance with a 256-channel array at 50 subsampled trials was higher than the other subjects with 128 channel arrays, indicating that improvements in decoding accuracy may be possible with increased channel counts. This improved decoding with coverage was specific to electrodes from SMC, and HG activations from inferior-frontal regions did not significantly contribute to the performance (Supplementary Fig. 11).

S2 displayed labial and dorsal-tongue clusters over the pre-central (motor) gyrus; and S4 demonstrated distinct articulatory clusters within pre-central gyrus, indicating that electrodes with high speech articulator information were contained within SMC.

7) Section 2.1: How accurately are grids coregistered with individual anatomy for the 3 128-grids slid through the burr hole? It seems unlikely that the 200 Micron electrodes could be distinguished on CT (no images shown). Why not use one of the localization techniques available (eg Gridlock, Branco *et al.*, Neuroimage 2018). Supplementary Fig 2 shows equal HG power in electrodes over sulci which deviates from papers with (more) accurate co-registration (eg the Branco paper)

In the first submission, we were not able to co-register the electrode arrays on the individual subject's brains due to limitations with obtaining sufficiently clean T1 weighted imaging data required for cortical reconstruction. Therefore, we manually localized the 128-channel grid and 256-channel grid over an average MNI brain based on the guidance from the neurosurgeon. However, in this revision, we have incorporated a recently developed pipeline¹⁹ to clean up noisy MR images, which enabled accurate brain reconstructions for each subject. The electrode markings obtained from intraoperatively recorded CT images or BrainLab Neuronavigation software, were then superimposed onto the reconstructed brain for subject-specific co-registration. We added Supplementary Fig. 2 that displays the reconstruction and localization results for each subject. We agree with the reviewer that individual 200-micron electrodes were unlikely to be distinguished on CT images, however, we were able to visualize the overall electrode array. We manually marked the corners of array locations (distal ends) from the CT scans (or BrainLab coordinates) for localization purposes.

We thank the reviewer for suggesting an unsupervised procedure (Branco *et al.*, 2018²⁰) to automatically co-register the grid using the underlying anatomical and vasculature information. Unfortunately, we were unable to accurately use this method, since we did not record angiogram information during the intra-operative procedure. However, as we continue to refine our localization approach, we will investigate using this procedure for future work. We would also like to emphasize that we have added a new method for individual brain reconstruction and localization, that we have now outlined in the manuscript:

Methods (Section 4.2)

In order to co-register the electrode arrays on the individual subject's brain, the array locations were first assessed using intraoperative CT scan (S1, S2, and S3) or registration markings from Brainlab Neuronavigation software (S4). For each subject, the cortical surface was reconstructed from a preoperative MRI image, using Freesurfer^{19,21}. For subjects S1, S2, and S3, the array landmarks (distal ends) were then localized using BiImage Suite, after aligning the reconstructed T1 volume with CT scans. For subject S4, we used key anatomical landmarks to localize the BrainLab coordinates on the individual subject's reconstructed brain. Finally, to localize individual electrodes, the electrode array templates (for both 128 & 256 channels) were then mapped to each individual subject brain by manually aligning the template to the array locations. **Fig. 1D** and Supplementary Fig. 2 show three templates of 128-channel grids (violet, green, and blue for S1, S2, and S3 respectively), and a 256-channel template (S4: purple) implanted over SMC of a subject averaged brain and subject-specific brains, respectively.

Discussion

While this spatial characterization resulted in anatomical maps on subject-specific brains, the individual electrode locations were obtained by manually overlaying the array templates on the markers from intra-operative imaging data. Future work will include designing μ ECoG grids with fiducial markers and developing automated co-registration techniques to more accurately localize μ ECoG array onto the individual subject's brain surface^{20,22}. Nevertheless, these subject-specific articulator maps indicate the potential of high-density μ ECoG for future neural prostheses for individual patients.

8) Section 2.2: Better decoding of P1 is most likely due to phoneme transitions in P2 and P3. In BCI applications most phonemes will be in transitional positions, especially when people do not pause between words. This limits the decoding performance 'in real life' and should be discussed.

We acknowledge the reviewer's point that the decoding performance for embedded phonemes (P2 and P3) are lower compared to the first-position phonemes. We think the reduced performance could also be due to discrepancies with marking voice onset times of embedded phonemes and co-articulatory effects that arise during phoneme transitions. We systematically investigated these effects in section 2.2 and section 2.4, where we found that optimal decoding was specific for each position and all phonemes (both the first position and embedded) can be sequentially decoded with respect to the utterance onset (thereby removing the necessity to measure the onset times of embedded phonemes for the decoder). Further, we developed a seq2seq decoding framework to decode all phoneme sequences simultaneously with respect to utterance onset. This sequential modeling will enable speech decoding that is not reliant on explicit phoneme onsets.

We included following text in the Discussion section:

Discussion:

The decoding performance decreased for phonemes embedded within the non-word, most likely due to discrepancies from marking onset times of embedded phonemes and co-articulatory effects that arise during phoneme transitions. To investigate these effects, we examined the amount of time necessary to decode each phoneme (Supplementary Fig. 12). We observed increased decoding performance with larger windows until saturation at around 500 ms similar to previous results^{2,4}. This optimal window was specific for each position indicating a need for decoding models that do not require explicit phoneme onsets. This decoding window selection, however, is essential for online systems with an optimal tradeoff between speed and accuracy¹⁸.

9) Section 2.2 and Fig 4C: tasks appear to differ between experiments given only 4 phonemes were included. Is it possible that phonemes of the ECoG and sEEG experiments were embedded and included phoneme transitions? (Hence the more accurate comparison should be average of P2 and P3)

Patients with SEEG implants underwent the same non-word repetition task as described in section 4.4, and we considered only vowels at P1. However, patients implanted with ECoG repeated instances of different stimulus tokens (heat, hot, hut, and hoot) which differed in vowel phonemes. /h/ being a voiceless glottal fricative, we theorized that the articulatory effects between the tokens should be strongly explained by the subsequent vowel. We, therefore compared the decoding performance of P1-vowels from μ ECoG and SEEG to the vowel decoding from ECoG.

10) Idem: the comparison in performance to clinical ECoG and sEEG is weak because in those subjects the SMC coverage is likely to be very sparse (especially sEEG) so the lesser performance is not very meaningful in that if you were to implant clinical scale grids for BCI the coverage would be much better.

We acknowledge the reviewer's concern about the limited coverage of clinical IEEG systems. We included Supplementary Fig. 16 that displays the location of SEEG and ECoG electrodes for each patient on an average MNI brain. Also, we calculated the approximate coverage of SEEG and ECoG in Supplementary Table 1, which indicates that the simulated coverage is comparable to our μ ECoG coverage. In addition, the decoding performances from our clinical ECoG (36.7%) and SEEG (36.5%) were comparable to the performance from existing studies using standard clinical ECoG (40.6%; 4-way vowel decoder; Pei *et al.*, 2011)²³. While it is true that decoding performance could be better for a clinical scale grid that is anatomically targeted BCI, we did not have access to this type of data and furthermore, our subsampling results (Fig. 5 and Supplementary Fig. 12) suggest that even with the same anatomical coverage, μ ECoG performs better.

We included Supplementary Fig. 16, that shows locations for IEEG electrodes.

11) Section 2.2 and Fig 5A: subsampling does not take into account that multiple electrodes are positioned over sulci. These do not contribute much to decoding including all electrodes but they can disproportionately contribute their noise to randomly subsampled configurations

We recognize the reviewer's concern with electrodes positioned over sulci that could potentially contribute noise to the decoding. We repeated the subsampling analysis from Fig. 5A with electrodes removed from the central sulcus. The analysis revealed that decoding performance with sulcus electrodes removed was comparable to decoding values at higher spatial resolutions. Our empirical results indicate that electrodes positioned over sulci did not significantly affect the decoding performance and the functional relationship between the decoding performance and electrode spacing. Unfortunately, we were to repeat the subsampling analysis with the sulcus electrode removed, that would test for spatial coverage and contact-size, as the subsampling would result in spatial grids with varying number of electrodes, thereby, compromising the fixed resolution condition.

We present the result as a reviewer figure below and can include to the manuscript based on the reviewer's preference.

Reviewer figure 1: Subsampling analysis on S1 with electrodes removed from central sulcus. Left: Electrode array map with electrodes removed from central sulcus (white indicates absence of electrodes); Right: Effect of spatial resolution on decoding performance. P1 – decoding accuracy was calculated for subsampled electrodes at maintained coverage with (blue) and without (green) electrodes from central sulcus. Performance for both conditions is comparable.

12) Supp Fig 7C: The graph seems to plateau after 3-4 classes. I would rephrase the claim that with number of classes even more than 9 micro-ECoG is likely to provide larger decoding benefits compared to less dense ECoG grids.

We thank the reviewer for the suggestion. We rephrased the claim in the supplementary figure (current Supplementary Fig. 14):

Spatial resolution required to achieve 90-95% of maximum decoding performance increased with increase in N and saturated to resolutions lower than 1.5 mm for N greater than 5.

MINOR

13) In Section 2.1 please explain the task better in the results section (info given in support table 2) and number of repetitions of nonsense words and phonemes

We included details about non-words and phonemes in the Results under Section 2.1.

Each of the subjects completed three task blocks (52 unique tokens per block; three repetitions overall) of a speech repetition task, during which the subjects were asked to listen to and repeat back auditorily presented non-words. Each non-word stimulus was either a CVC or VCV token, with a fixed set of 9 phonemes (4 vowels and 5 consonants) at each position within the token (see Methods).

14) Line 105 and Fig 1: grid dimensions do not agree: fig 1: 12x22, line 105 12x24, both '256 channels;'

We have corrected the grid dimensions to 12x22.

15) Line 126: the authors state that the increase in high gamma aligned to speech onset (-250 ms to 250 ms around speech onset), however Figure 2B (and possibly 2A) seems to show increases from -1 second prior to speech onset. Can the authors clarify this?

We acknowledge the reviewer's comment and modified the text in Results (under section 2.1) to the following:

HG power increases were aligned to the speech utterance in individual electrodes and were active up to 1000 ms before utterance onset, **Fig. 2B**, Supplementary Fig. 4) and were identified as statistically significant as compared to a pre-stimulus baseline using a non-parametric permutation test with an FDR-corrected alpha threshold of $p < 0.05$ (see Methods). Significant electrodes are highlighted in Figure 2 with black borders: S1 111/128 significant channels, S2 111/128, S3 60/128, and S4 149/256. These electrodes exhibited spatially varying characteristics of HG activations with 77.4% of these electrodes (S1 107/111, S2 107/111 S3 34/63, and S4 88/149) were active before the utterance start, indicating that μ ECoG electrodes measured earlier motoric activations leading to speech (Supplementary Fig. 5).

16) Line 142: Is the correlation calculated on baseline data or task data? This matters in that the latter should yield high correlations by default

We calculated the correlation on task data with respect to speech onset. While task data correlations are likely to be higher than the baseline, we were interested in quantifying the relative signal-sharing of speech signals at different spatial resolutions.

We made the following correction to the Results under section 2.1:

To quantify the spatial resolution of μ ECoG neural signals, we computed the Pearson correlation coefficient of the HG envelope (-500 ms to 500 ms with respect to speech utterance onset) between each micro-electrode pair.

17) Line 184: Could the authors specify how many principal components (that altogether explain 80% variance) were used for decoding?

We added the equivalent number of principal components in the Results (under section 2.2) that explained 80% variance:

We selected the eigenvalues based on nested cross-validation that explained 80% of neural variance (equivalent number of principal components: S1 - 34, S2 - 39, S3 - 25, S4 - 22) for phoneme prediction.

18) Line 186: Please explain what 'position' means in line 186

We explained the term position in the sentence and provided an example non-word token in the Results under section 2.2.

Benefiting from high spatial sampling, we observed strong decoding performance in all subjects for predicting phonemes in all positions within the non-word (e.g., /abae/: P1 - /a/, P2 - /b/, P3 - /ae/; **Fig. 4A & 4 B**, $p < 0.01$, Binomial test against chance model).

19) Line 332: Can the authors elaborate on why they used a bidirectional RNN over a unidirectional one? For a real-time system with a BCI user, a unidirectional RNN may be preferred. Besides, what would be the rationale for expecting the information about the present phoneme to be in neural signals almost a second later?

We motivated the architecture based on the previously established neural machine-translation models that employed bidirectional RNNs for the encoder²⁴. Compared to the unidirectional model, the bidirectional RNN can enable stable encoding of phoneme sequences by linking neural information in each time bin from both directions (forward and reverse modeling). We acknowledge the reviewer's concern with real-time BCI systems, where information from the future will not be available for reverse modeling. However, we predict our BCI systems will operate over a time-window of neural recordings (as opposed to every time point), within which information can be modeled in both directions. Further, these bidirectional RNNs (in the form of gated-recurrent-units) have been employed in real-time speech BCI studies that resulted in successful word predictions from an anarthric patient¹⁰.

We recognize the reviewer's concern on information about earlier phonemes (time bin 1) not being present at later times (time bin N). However, we would like to emphasize that we wanted to design a sequential model that retrieves all the phoneme content as it parses neural data through time. To achieve this, the encoder RNNs (in the form of LSTMs) contain hidden cells to sequentially model information along the time-bins resulting in a fixed length vector as a final encoder state. This final encoder state should contain not only the **phoneme set**, but also the information about the **ordering of phonemes**. Finally, the decoder LSTMs were used to retrieve these phonemes from the final encoder state, in a sequential manner.

20) Line 431: It looks like the authors did not use any statistical information about phoneme sequences in their RNN. Please make this more explicit in the results section as well.

We addressed the comment by adding information on phoneme statistics in section 2.5.

We developed a phoneme generation technique to sequentially decode positional phonemes within the utterance, without prior knowledge on the statistical relationship between phonemes (phonotactics).

21) Fig 2A, B: are data from averaged trials?

We averaged the spectrograms across all spoken trials. We included information about averaging in the figure 2 caption and section 2.1.

Figure 2 caption:

Spectrograms for each electrode from each array for two example subjects (averaged across all spoken trials): S1, 128 electrodes (1.33 mm spacing) & S4, 256 electrodes (1.72 mm spacing).

Section 2.1:

On examining speech neural activations, we observed significant modulation of spectro-temporal neural activations (multi-taper spectrogram estimate averaged across all spoken trials, see Methods) during speech articulation, including prominent HG band power increases.

22) Supp Table 4: What coordinate system is used for X,Y coordinates?

We used a coordinate system with the vertical dimension as x-coordinate, the horizontal dimension as y-coordinate, and the top-left corner of the array as the origin. We added the coordinate system to the footnote of Supplementary Table 4.

*top-left corner as origin

Reviewer #2 (Remarks to the Author):

The authors investigate how speech decoding can be improved with μ ECoG electrodes that have a higher spatial resolution compared to standard and high-density ECoG grids. In the present study, the authors conducted speech production tasks with four patients who were implanted with μ ECoG grids to identify phonemes in the neural data via linear (LDA) and non-linear (Neural Network) classification techniques. Decoding results are compared to those acquired from ECoG grids with 4mm and 10mm inter-electrode spacing, as well as to stereotactic EEG. The authors give a comprehensive analysis in which aspects μ ECoG grids might provide superior information for speech decoding.

This is a strong paper, and the results show significant advances to the field. Overall, the manuscript is well-written, experimental details are clearly presented, the data analysis is properly documented, and all figures are well embedded into the main text and further underline their claims. The work presented in this manuscript is original and meets the expected standards in the field. I only have one moderate and a few minor comments / questions before the manuscript is ready to be published.

We thank the reviewer for their support of our work and providing detailed reviews and feedback on our manuscript. Responses to comments and revisions are addressed point-by-point, below.

In section 2.1 the authors describe the subject pool and which tasks they conducted. In line 116, it is not clear to what part in the experiment design the term “utterance” refers to. Please clarify with respect to the experiment tasks how (average) utterance duration differs from repeating the auditorily presented words. Maybe the term can also be visualized in Figure 1E.

We acknowledge the reviewer’s point on the ambiguity between “utterance” and “repeating the auditorily presented words”. We define “utterance” as the participant’s spoken part of the speech repetition task. We made the following changes in Results under section 2.1 and figure 1E to address this comment.

Section 2.1: Subjects took on average 1.1 seconds (range = 0.7 to 1.5 s) to repeat auditorily presented non-words and had an average spoken utterance duration of 450 ms (range = 300 to 700 ms) (**Fig. 1E**, Supplementary Table 3).

Figure 1E: A schematic of the intraoperative speech production task. Color bars indicate the duration of the auditory stimulus (blue), time-to-response (orange), and spoken utterance duration (green).

In section 2.1 the authors investigate how articulatory features are organized in the neural space. I suggest adding one sentence to the main text describing how the articulatory features were inferred. This information is a bit hidden in the Methods section, and in section 2.2 about the phonological distances.

We acknowledge the reviewer’s comment about missing information on how articulatory features were inferred. We grouped phoneme labels into 4 articulators: low-vowel - /a/, /ae/, high-vowel - /i/, /u/, labial-consonant - /b/, /p/, /v/, and dorsal tongue consonant - /g/, /k/.

We made the following changes in Results section 2.1 and section 2.2

Section 2.1: To investigate micro-scale neural information, we examined spatio-temporal activations specific to key units of speech production: articulatory features of individual phonemes (low-vowel - /a/, /ae/, high-vowel - /i/, /u/, labial-consonant - /b/, /p/, /v/, and dorsal tongue consonant - /g/, /k/).

Section 2.2: We reasoned that if our ability to decode phonemes was largely based on our ability to resolve their constituent articulatory features in the motor cortex, decoding errors should not be uniform across phonemes, but rather reflect similar articulatory properties (low, high, labials, and dorsals) from other phonemes.

In Figure 2A, the authors visualize spectral activity from each electrode for subjects S1 and S4. Based on the main text, it is not clear to me what exactly is represented in the spectrograms: Is it the average activity of a particular nonword token, or representing HG of speech activity in general? I suggest adding one sentence clarifying that.

We apologize for the confusion. Each electrode in figure 2A depicts an average spectrogram estimate across all spoken trials.

We made the following changes in Results under section 2.1 and figure 2A (captions):

Section 2.1: We observed significant modulation of spectro-temporal neural activations (multi-taper spectrogram estimate averaged across all spoken trials, see Methods) during speech articulation, including prominent HG band power increases.

Figure 2A: Spectrograms for each electrode from each array for two example subjects (averaged across all spoken trials): S1, 128 electrodes (1.33 mm spacing) & S4, 256 electrodes (1.72 mm spacing).

In Figure 2C, the authors mention dotted lines. However, in the figure there are no dotted lines, and I believe they refer to the solid ones.

We apologize for the error and have corrected the caption:

The red lines and the blue boxes indicate the median and 25/75th percentile and solid lines represent the full-range of the distribution.

In section 2.1, the authors use poisson disc sampling to examine the contributions of the higher spatial resolution by repeating the state-space analysis on a subsampled spatial resolution. I am not fully convinced if the poisson disc sampling method is appropriate here as it might result in electrode clusters which are spatially aligned in a closer distance to what is capable of standard or high-density ECoG (as indicated by Fig. 4S). To my understanding, the r-value (radius) from the poisson disc sampling method should be chosen to be either 4mm or 10mm to provide comparable results to currently used grids, but this would also enable much lower spatial resolutions as poisson disc sampling uses the spherical annulus between r and 2r for selecting the next electrode, and that could bias the analysis in favor of the proposed method. Happy to stand corrected on this ground.

We selected Poisson disc sampling to preserve the spatial homogeneity in selecting electrodes during subsampling. Contrary to the selected method, a random subsampling can result in non-uniform clustering of electrodes leading to a high variance in analysis. We acknowledge the author's point that selecting the r-value in advance, could indeed bias the analysis in favor of the proposed method. Therefore, we used the r-value from Poisson disc sampling only as a guide to selecting the number of electrodes and evaluated the performance as a function of the number of electrodes subsampled.

In section 2.1, we wanted to demonstrate that a decrease in spatial subsampling (with uniform distribution across the array) would result in degraded performance. For each subsampling instance, we were interested in the total number of electrodes that we obtained by parametrically varying the r-value of the Poisson disc sampling (E.g., 64 uniformly subsampled electrodes from a 128-electrode array would result in 50% subsampling). To mitigate the effect of spatial clusters during each subsampling, we performed a Monte-Carlo analysis over 50 instances of similar subsampling for each group in order to estimate the performance distribution.

Correspondingly, in section 2.3, on studying the effect of spatial resolution on decoding performance, we calculated the pitch or inter-electrode distance in a post-hoc manner as a function of the number of subsampled electrodes (formula 1), which indeed can be different from the Poisson-disc radius used for subsampling. This post hoc version of calculating the inter-electrode distance would provide comparable results to clinical ECoG grids.

In section 2.2, please clarify in the main text (one sentence should be sufficient) what you mean with all positions, position one, position two and position three to avoid ambiguity. It is not clear from the main text that it refers to the position within the nonword of the CVC or VCV.

We acknowledge the reviewer's comment with ambiguity in describing phoneme positions. We made the following changes to Results under section 2.2.

Section 2.2: Benefiting from high spatial sampling, we observed strong decoding performance in all subjects for predicting phonemes in all positions within the non-word (e.g., /abae/: P1 - /a/, P2 - /b/, P3 - /ae/; Fig. 4A & 4 B, $p < 0.01$, Binomial test against chance model).

Language: The manuscript uses the term μ ECoG extensively in the first part, but switches to high-density μ ECoG for the second (and then back). I suggest keeping it consistent. I also detected a few minor typos: ... "my" median decoding accuracy in the caption of Figure 5S, "um" diameter in caption of Figure 1.

We changed the manuscript to use "high-density μ ECoG" to describe the whole electrode array and changed all instances of μ ECoG to high-density μ ECoG. We use the term " μ ECoG electrode" to describe an individual electrode. We have corrected the typos identified by the reviewer.

Reviewer #3 (Remarks to the Author):

In "High-resolution neural recordings improve the accuracy of speech decoding", Duraivel et al. compared μ ECoG recordings with ultra-high density spatial resolution in four participants to standard ECoG/sEEG recordings to determine whether and how decoding speech information might benefit from increased spatial resolution afforded by μ ECoG. Overall, I think the authors have convincingly shown the benefits of using higher resolution arrays for speech decoding and for resolving spatiotemporal neural activity in general. I also appreciate the analyses comparing both downsampled or spatially smoothed versions of the μ ECoG array data to typical sEEG and ECoG, as this provides additional context for why higher density ECoG may result in better information for both encoding and decoding analyses. I have a few major and minor comments that I believe will improve the manuscript in its current form.

We thank the reviewer for their support of our work and providing detailed reviews and feedback on our manuscript. Responses to comments and revisions are addressed point-by-point, below.

Major Comments:

1. The authors used a nonword task for their repetition task in the patients, but it is not explained why this was chosen over a word repetition or other closed vocabulary task, as is used for typical speech decoding experiments. Some additional text regarding the rationale behind this choice would be helpful.

We acknowledge the reviewer's concern about using a non-word task for speech repetition. For this study, we are interested in explicitly decoding speech articulatory features and corresponding phonemes to demonstrate the utility and benefit of high-density μ ECoG arrays. We specifically designed this non-word task to remove the effects of semantic and higher-order language features on our decoding to fully assess our ability to resolve speech motor features. Further, we controlled for a fixed set of phonemes per position within the non-word stimulus which allowed us to explicitly and systematically examine speech phoneme articulators at different positions within the non-word.

We included following sections to the Discussion:

Discussion

We performed speech decoding by classifying phonemes from all positions within the spoken non-word. Using non-word stimuli in the speech task enabled us to remove the effects of higher-order language processes and to decode phonemes and articulatory features to clearly assess the benefits of using high-density μ ECoG to decode speech features from neural recordings.

2. Information about the macro ECoG/sEEG recordings and the placement of arrays is needed. At the moment, only a text description saying that the electrodes were placed in SMC or related areas is mentioned, but it is not possible to directly compare the placement of these electrodes to the placement of the μ ECoG grid. With μ ECoG grids, if the overall array size

is smaller, it will be necessary to choose a more narrowly defined anatomical region from which to record. As seen in Figure 2 and Supplementary Figure 2, the arrays cover different anatomical regions of interest, with many of the strongest responses arising from postcentral and precentral gyrus. For sEEG, it is unclear how similar responses might look and whether they are sampling from the same region. Although I appreciate that these arrays should have higher resolution than most sEEG of this area, having a more direct comparison of the anatomy would be helpful.

We added Supplementary Fig. 16 with locations of SEEG and ECoG electrodes over an average MNI brain (color coded with respect to subject). Our IEEG electrodes recorded signals from both post-central and pre-central gyrus, which is similar to the coverage of our μ ECoG arrays (Supplementary Fig. 16). In addition, the decoding performances from our clinical ECoG (36.7%) and SEEG (36.5%) were comparable to the performance metrics from existing studies using standard clinical ECoG (40.6%; 4-way vowel decoder; Pei *et al.*, 2011)²³. Further, we provided the estimate of area coverage from SEEG in Supplementary Table 1 (Clinical summary of patients: pre-operative epilepsy monitoring).

3. In Figure 3 and Supplementary Figure 3, the authors show the spatiotemporal high gamma across the arrays in all four subjects with μ ECoG, and find a striking difference in patterns. How does this pattern compare to known topography of jaw, larynx, tongue, etc. as shown by, e.g. Bouchard *et al.* Nature 2013? While this paper and others are cited, it is not explained whether their current results show similar somatotopy or not.

As the reviewer mentioned, we observed striking differences in spatial patterns with respect to phoneme onsets. To address the reviewer's comments, we added supplementary figures 5 and 6 in which we spatially laid out the features of HG activations for all subjects. To show anatomical features, we overlaid electrode array location onto the pre-central, central, and post-central sulcus for each individual subject's brains (supplementary figures 4, 5, 6, 11, and 15) and added to commentary reflecting this addition to the Discussion section (see below).

Further, to the reviewer's point, we observed similarities in findings between the topographies revealed by Bouchard *et al.* (Nature 2013)²⁵ and the articulator maps for S1 (Fig. 5B). We observed clustering of dorsal (tongue) articulators in the anterior-inferior region and clustering of labials (lips) in the posterior-superior region of SMC (analogous to blue and red markers; Figure 2, Bouchard *et al.* Nature 2013).

We included the following text in our Discussion.

Discussion

Even with the time-constrained experimental procedure (up to 15 minutes of recording duration/ 1.04 minutes of total spoken duration; **Fig. 1E**, Supplementary Table 3), our μ ECoG electrodes recorded neural signals in SMC that produced significantly higher HG-ESNR as compared to standard intracranial methods (ECoG/SEEG). Electrodes with HG activations spanned both the pre-central (motor) and post-central (sensory) gyrus across the SMC for S1 (Supplementary Fig. 2 & 4). Subject S2 and S3 had predominant coverage in the pre-central and post-central gyrus, respectively. Subject S4 had a 256-channel array that extended anteriorly from SMC to the inferior frontal gyrus, however, most HG activations were confined to SMC.

On a subject-specific scale, we observed high-resolution speech articulator maps, similar to previously established group-level organization¹⁶. We observed that the surface area for each articulator map ranged from 11 mm² to 80 mm² depending on the location and coverage of each μ ECoG grid (Supplementary Table 4). Overlaying these spatial clusters on each subject's individual brain, we made the following anatomical observations: S1 and S3 exhibited labial clusters near the posterior region of post-central (sensory) gyrus, and dorsal-tongue clusters (only S1) near the anterior-inferior region. S2 displayed labial and dorsal-tongue clusters over the pre-central (motor) gyrus; and S4 demonstrated distinct articulatory clusters within pre-central gyrus, indicating that electrodes with high speech articulator information were contained within SMC. This anatomical clustering is similar to previously identified somatotopic representations, that revealed a posterior clustering of lip articulators and an anterior clustering of tongue articulators with respect to central sulcus¹⁶.

Minor Comments:

1. In Figure 3, the authors show the difference between spatiotemporal activity for four different phonemes, which is further expanded in Supplementary Figure 3. It would be helpful for the authors to point out some of the more subtle differences

that can be observed between phonemes that differ by only one feature, for example, /b/ vs. /p/, which differ by voicing, and there does seem to be a spatial pattern of activity in /b/ that overlaps somewhat with voiced compared to voiceless phonemes.

To the reviewer's point, the voiced phonemes within a speech articulator (/b/ in labial, /g/ in dorsal) exhibited increased HG power as compared to voiceless counterparts (/p/ in labial, /k/ in dorsal).

We made the following changes to the figure captions for the current Supplementary Fig. 7 (earlier version: Supplementary Fig. 3)

Subtle differences in spatial patterns of activation and relative HG power can be observed between phonemes that differ by one articulatory feature (/b/ vs. /p/, and /g/ vs. /k/).

2. For Supplementary Figure 2, I appreciate seeing the diversity of spectrotemporal information afforded by using μ ECoG grids – the differences are quite striking. However, I do wonder why the MNI brain was chosen for viewing, since there are only four participants, and it would be more accurate to show the arrays on the native brain.

In the first submission, we were not able to co-register the electrode arrays on the individual subject's brains due to limitations with obtaining sufficiently clean T1 weighted imaging data required for cortical reconstruction. Therefore, we manually localized the 128-channel grid and 256-channel grid over an average MNI brain based on the guidance from the neurosurgeon. However, in this revision, we have incorporated a recently developed pipeline¹⁹ to clean up noisy MR images, which enabled accurate brain reconstructions for each subject. The electrode markings obtained from intraoperatively recorded CT images or BrainLab Neuronavigation software, were then superimposed onto the reconstructed brain for subject-specific co-registration. We added Supplementary Fig. 2 that displays the reconstruction and localization results for each subject. We agree with the reviewer that individual 200-micron electrodes were unlikely to be distinguished on CT images, however, we were able to visualize the overall electrode array. We manually marked the corners of array locations (distal ends) from the CT scans (or BrainLab coordinates) for localization purposes. We then used the design map of the electrode arrays to localize individual contacts. Future work will focus on developing better tools to localize individual electrode contacts.

We included the details of anatomical reconstruction in the Methods section and the limitation on manual co-registration of our array in our Discussion section.

Methods (Section 4.2)

To co-register the electrode arrays on the individual subject's brain, the array locations were first assessed using intraoperative CT scan (S1, S2, and S3) or registration markings from Brainlab Neuronavigation software (S4). For each subject, the cortical surface was reconstructed from a preoperative MRI image, using Freesurfer^{19,21}. For subjects S1, S2, and S3, the array landmarks (distal ends) were then localized using BiImage Suite, after aligning the reconstructed T1 volume with CT scans. For subject S4, we used key anatomical landmarks to localize the BrainLab coordinates on the individual subject's reconstructed brain. Finally, to localize individual electrodes, the electrode array templates (for both 128 & 256 channels) were then mapped to each individual subject brain by manually aligning the template to the array locations. **Fig. 1D** and Supplementary Fig. 2 show three templates of 128-channel grids (violet, green, and blue for S1, S2, and S3 respectively), and a 256-channel template (S4: purple) implanted over SMC of a subject averaged brain and subject-specific brains, respectively.

Discussion

While this spatial characterization resulted in anatomical maps on subject-specific brains, the individual electrode locations were obtained by manually overlaying the array templates on the markers from intra-operative imaging data. Future work will include designing μ ECoG grids with fiducial markers and developing automated co-registration techniques to more accurately localize μ ECoG array onto the individual subject's brain surface^{20,22}.

3. For Supplementary Figure 3, why is there a * and # after S3 and S4?

We included the symbols * and # to indicate S3 had low-SNR recording and S4 completed 1 out of 3 blocks of the speech repetition task.

We made the following changes to the figure captions for Supplementary Fig. 7 (earlier version Supplementary Fig. 3).

S3 and S4 exhibited weaker yet significant activations due to low SNR (*) and recording duration (#), respectively (* - low SNR recording, # - completed 1 block only).

4. Please consider enlarging font sizes for the smallest fonts for Figures 1-6. Especially, for example, the legend in Fig. 6C and the axis labels in 6D are difficult to read at the current size.

We have increased the font sizes for the above-mentioned figures.

1. Kellis, S. *et al.* Decoding spoken words using local field potentials recorded from the cortical surface. *J. Neural Eng.* **7**, 056007 (2010).
2. Mugler, E. M. *et al.* Direct classification of all American English phonemes using signals from functional speech motor cortex. *J. Neural Eng.* **11**, 035015 (2014).
3. Ramsey, N. F. *et al.* Decoding spoken phonemes from sensorimotor cortex with high-density ECoG grids. *Neuroimage* **180**, 301–311 (2018).
4. Wilson, G. H. *et al.* Decoding spoken English from intracortical electrode arrays in dorsal precentral gyrus. *J. Neural Eng.* **17**, 66007 (2020).
5. Livezey, J. A., Bouchard, K. E. & Chang, E. F. Deep learning as a tool for neural data analysis: Speech classification and cross-frequency coupling in human sensorimotor cortex. *PLOS Comput. Biol.* **15**, e1007091 (2019).
6. Herff, C. *et al.* Brain-to-text: decoding spoken phrases from phone representations in the brain. *Front. Neurosci.* **9**, 217 (2015).
7. Williamson, J. H., Quek, M., Popescu, I., Ramsay, A. & Murray-Smith, R. Efficient human-machine control with asymmetric marginal reliability input devices. *PLoS One* **15**, e0233603 (2020).
8. Shoham, S., Halgren, E., Maynard, E. M. & Normann, R. A. Motor-cortical activity in tetraplegics. *Nature* **413**, 793 (2001).
9. Hochberg, L. R. *et al.* Neuronal ensemble control of prosthetic devices by a human with tetraplegia. *Nature* **442**, 164–171 (2006).
10. Moses, D. A. *et al.* Neuroprosthesis for decoding speech in a paralyzed person with anarthria. *N. Engl. J. Med.* **385**, 217–227 (2021).
11. Anumanchipalli, G. K., Chartier, J. & Chang, E. F. Speech synthesis from neural decoding of spoken sentences. *Nature* **568**, 493–498 (2019).
12. Giraud, A.-L. & Poeppel, D. Cortical oscillations and speech processing: emerging computational principles and operations. *Nat. Neurosci.* **15**, 511–517 (2012).
13. Zoefel, B. Speech entrainment: Rhythmic predictions carried by neural oscillations. *Curr. Biol.* **28**, R1102–R1104 (2018).
14. Ding, N., Melloni, L., Zhang, H., Tian, X. & Poeppel, D. Cortical tracking of hierarchical linguistic structures in connected speech. *Nat. Neurosci.* **19**, 158–164 (2016).
15. Oganian, Y. *et al.* Phase Alignment of Low-Frequency Neural Activity to the Amplitude Envelope of Speech Reflects Evoked Responses to Acoustic Edges, Not Oscillatory Entrainment. *J. Neurosci.* **43**, 3909 LP – 3921 (2023).
16. Chartier, J., Anumanchipalli, G. K., Johnson, K. & Chang, E. F. Encoding of articulatory kinematic trajectories in human speech sensorimotor cortex. *Neuron* **98**, 1042–1054 (2018).
17. Roussel, P. *et al.* Observation and assessment of acoustic contamination of electrophysiological brain signals during speech production and sound perception. *J. Neural Eng.* **17**, 56028 (2020).
18. Shenoy, K. V., Willett, F. R., Nuyujukian, P. & Henderson, J. M. *Performance Considerations for General-Purpose Typing BCIs, Including the Handwriting BCI.*
19. Iglesias, J. E. *et al.* SynthSR: A public AI tool to turn heterogeneous clinical brain scans into high-resolution T1-weighted images for 3D morphometry. *Sci. Adv.* **9**, eadd3607 (2023).
20. Branco, M. P., Leibbrand, M., Vansteensel, M. J., Freudenburg, Z. V & Ramsey, N. F. GridLoc: An automatic and unsupervised localization method for high-density ECoG grids. *Neuroimage* **179**, 225–234 (2018).
21. Dale, A. M., Fischl, B. & Sereno, M. I. Cortical surface-based analysis: I. Segmentation and surface reconstruction. *Neuroimage* **9**, 179–194 (1999).
22. Gupta, D., Hill, N. J., Adamo, M. A., Ritaccio, A. & Schalk, G. Localizing ECoG electrodes on the cortical anatomy without post-implantation imaging. *NeuroImage Clin.* **6**, 64–76 (2014).
23. Pei, X., Barbour, D. L., Leuthardt, E. C. & Schalk, G. Decoding vowels and consonants in spoken and imagined

- words using electrocorticographic signals in humans. *J. Neural Eng.* **8**, 46028 (2011).
24. Makin, J. G., Moses, D. A. & Chang, E. F. Machine translation of cortical activity to text with an encoder–decoder framework. *Nat. Neurosci.* **23**, 575–582 (2020).
 25. Bouchard, K. E., Mesgarani, N., Johnson, K. & Chang, E. F. Functional organization of human sensorimotor cortex for speech articulation. *Nature* **495**, 327–32 (2013).

REVIEWER COMMENTS

Reviewer #1 (Remarks to the Author):

Overall my concerns have been addressed, for which i thank the authors. Yet, one issue still concerns me, addressed in comment nr 10 which i may not have explained adequately in my first review. Crux of the manuscript is comparison to lower density electrodes, but the comparison is not optimal given the depth electrodes with 4 mm spacing cannot utilize the topographic organization of the sensorimotor cortex. Typically only the electrodes that sample from the cortical grey matter layer records decodable information. Being depth electrodes, that will typically be only one electrode per lead (lead having multiple electrodes along its length). Hence, comparison addressing spatial coverage is not helpful. Rather, a fair comparison would be to define spatial resolution for this purpose as the distance between different depth leads where each penetrates the cortex surface, this not the distance between electrodes along one lead. Second, the clinical grids are not be placed optimally, also complicating comparison. Clinical grids are able to cover a large portion of SMC, thus are able to capture complimentary decodable information from cortical patches spaced apart, and yielding higher decoding performance (as the authors already show in their 'subsample grid size' analysis for the micro grids. To equal such coverage, the Micro ECoG grids would need to be expanded to maybe over 500 electrodes.

I would like to see these limitations addressed in the discussion, given that there are already impressive decoding results reported for grids with 3-4 mm pitch (with larger SMC coverage), which were probably not available thus not included in this manuscript. Thus, in my opinion, the jury is still out on whether decoding performance is better for the microgrids versus 3-4 mm pitch grids. Granted, the subsampling strategy gives some idea that microgrids are better, but there the subsamples contain considerably less electrodes, whereas the coverage of 3-4mm grids can be larger without requiring hundreds of amplifiers and likely produces better performance than the current sets of subsamples give.

I do believe higher density yield better results, and i think the authors did a very good job, but it seems to me that readers will easily assume that a comparison was made with densities used for BCI already (3-4 mm pitch grids). The authors mention that 'typical' ECoG recordings involve 4-10 mm pitch (line 65), and then refer to 10 mm grids and 4 mm sEEG leads as 'standard clinical'. Thinking along, perhaps this confusion can be prevented by discussing in terms of the benefit of micro ECoG *within a confined patch of cortex* (an activity hotspot), where subsampling is logical for comparison of densities, and separate this from the issue of covering multiple hotspots (where larger-pitch grids have a benefit). From there, one can explain that multiple micro grids can be placed on multiple hotspots to maximize the benefit, eg by using fMRI for guidance, and thereby outperform lower density grids. But I think that it should be mentioned much more clearly that the 4 mm pitch on sEEG leads does not compare to 4 mm pitch grids, since it is confusing. Come to think of it, the confusion i perceive would be gone if the sEEG data was not included in the manuscript.

Reviewer #2 (Remarks to the Author):

The authors have addressed all of my comments and clarified all my concerns. I have one minor suggestion regarding a change that was added in the revised version: Supplementary Figure 3 has 4 empty panels (I-L) indicating electrode-specific analysis regarding the absence of contamination in the HG band across subjects. I do not think that any of these panels add any additional value and suggest describing the absence purely in the text/caption.

Reviewer #3 (Remarks to the Author):

This paper assesses the contribution of μ ECoG recordings to improved decoding of speech from intracranial signals in sensorimotor cortex. The authors have adequately addressed my concerns in this revision, and have included important methodological details and clarifications, as well as several figures with information about localization and specific contrasts of interest. I have no further concerns.

Overall my concerns have been addressed, for which I thank the authors. Yet, one issue still concerns me, addressed in comment nr 10 which I may not have explained adequately in my first review. Crux of the manuscript is comparison to lower density electrodes, but the comparison is not optimal given the depth electrodes with 4 mm spacing cannot utilize the topographic organization of the sensorimotor cortex. Typically only the electrodes that sample from the cortical grey matter layer records decodable information. Being depth electrodes, that will typically be only one electrode per lead (lead having multiple electrodes along its length). Hence, comparison addressing spatial coverage is not helpful. Rather, a fair comparison would be to define spatial resolution for this purpose as the distance between different depth leads where each penetrates the cortex surface, this not the distance between electrodes along one lead. Second, the clinical grids are not placed optimally, also complicating comparison. Clinical grids are able to cover a large portion of SMC, thus are able to capture complimentary decodable information from cortical patches spaced apart, and yielding higher decoding performance (as the authors already show in their 'subsample grid size' analysis for the micro grids. To equal such coverage, the Micro ECoG grids would need to be expanded to maybe over 500 electrodes.

I would like to see these limitations addressed in the discussion, given that there are already impressive decoding results reported for grids with 3-4 mm pitch (with larger SMC coverage), which were probably not available thus not included in this manuscript. Thus, in my opinion, the jury is still out on whether decoding performance is better for the microgrids versus 3-4 mm pitch grids. Granted, the subsampling strategy gives some idea that microgrids are better, but there the subsamples contain considerably less electrodes, whereas the coverage of 3-4mm grids can be larger without requiring hundreds of amplifiers and likely produces better performance than the current sets of subsamples give.

I do believe higher density yield better results, and I think the authors did a very good job, but it seems to me that readers will easily assume that a comparison was made with densities used for BCI already (3-4 mm pitch grids). The authors mention that 'typical' ECoG recordings involve 4-10 mm pitch (line 65), and then refer to 10 mm grids and 4 mm sEEG leads as 'standard clinical'. Thinking along, perhaps this confusion can be prevented by discussing in terms of the benefit of micro ECoG *within a confined patch of cortex* (an activity hotspot), where subsampling is logical for comparison of densities, and separate this from the issue of covering multiple hotspots (where larger-pitch grids have a benefit). From there, one can explain that multiple micro grids can be placed on multiple hotspots to maximize the benefit, eg by using fMRI for guidance, and thereby outperform lower density grids. But I think that it should be mentioned much more clearly that the 4 mm pitch on sEEG leads does not compare to 4 mm pitch grids, since it is confusing. Come to think of it, the confusion I perceive would be gone if the sEEG data was not included in the manuscript.

We thank the reviewer for acknowledging our response and we accept the reviewer's concern with the lack of adequate information involving comparison of our high-density, high channel count μ ECoG electrodes against the lower density counterparts. To the reviewer's point, a SEEG shank with depth electrodes cannot completely characterize the topographic organization of the sensorimotor cortex, and the decoding ability can be limited to few electrodes per lead. However, this characterization concurs with the recording feature of SEEG technology, that can simultaneously record from cortex and deep brain regions along the shank but does not enable wider cortical coverage. Still, recordings from these implantations have been shown to accurately decode spoken features¹⁻⁴. A fair comparison would require multiple implantations of SEEG shanks over sensorimotor cortex, which in turn could complicate the surgical procedure. Further, the distance between SEEG contacts within gray matter across depths would mostly depend on anatomical considerations related to the surgical procedure itself (vasculature, optimal trajectory, etc.) as well as physical limitations of the implantation device (bolt size), as opposed to any technical limitations of the recording devices themselves. Consequently, an objective measure of distance between contacts in gray matter would not be as useful as it would be for standard ECoG. Nonetheless, we agree with the reviewer's concerns about the inherent limitations of SEEG sampling of gray matter. We also, we want to stress that the decoding performance that we obtained using SEEG is comparable to previously existing decoding results, indicating that our recordings were not systemically biased⁵. Therefore, we conclude our comparison results by contrasting the decoding performances between two recording technologies, of which higher spatial sampling is an essential but not the only factor.

Towards comparison against clinical grids (4 – 10 mm), we agree with the reviewer that clinical grids or macro-ECoG arrays (in our terminology) can cover larger portion of SMC when targeted optimally. Speech decoding studies with optimal targeting, as the reviewer mentioned, resulted in robust decoding performance^{6,7}. However, even with this optimal targeting and larger coverage, only small sections of the clinical grid sampled SMC, within which the decoding performance was driven by spatially selective cortical regions (hotspots, as mentioned by the reviewer). For example, in a study (Moses *et al.*, 2021)⁷ to decode attempted speech from an anarthric patient, the authors targeted SMC using a rectangular grid (65 mm long and 35 mm wide) with 4 mm inter-electrode distance. Within this coverage, the cortical sampling of SMC spanned (approximately) 55 mm long (dorsal-ventral) and 16 mm wide (anterior-posterior), and the highest cortical contributions were limited to 3 x 3 grids within these spanned regions. Similarly, we observed high-resolution speech articulatory cortical patches on a patient-specific scale (Fig. 5b and Supplementary Fig. 15). We argue that our study used micro-scale electrodes to sample these cortical patches up to 9x higher spatial resolution (Fig. 1b & 1c – note in particular the spatial density comparison to other ECoG technology in Figure 1b) and we demonstrated that highest decoding is maintained at less than 2 mm spatial resolution. We agree with the reviewer that the extra decoding performance could also be resolved with added coverage, and with this concern in mind, we designed the 12 x 22 array (S4), with the idea to use the longest side (28 cm long) to span SMC along dorsal-ventral orientation. However, the intraoperative implantation of μ ECoG arrays (for S4; Supplementary Fig. 2) were determined based on surgical considerations during tumor resection, resulting in the indicated spanning of SMC in Supplementary figure 2. To the reviewer's point, our next-generation electrode designs will span SMC along the dorsal orientation (3 x 6 cm) with the same high spatial resolution as our current design (1.3 mm pitch).

Finally, we acknowledge the reviewer's concern with 4-mm ECoG grids that are likely to be confused with SEEG electrodes with 4-mm inter-electrode spacing. With the phrase "typical ECoG recordings involve 4-10 mm pitch", we intend to refer studies using ECoG grids with surface cortical electrodes that are spaced 4 mm and 10 mm apart, respectively. We used the term "SEEG" to refer to depth electrodes. Collectively, we used the terms IEEG and standard clinical recordings to refer to both recording systems combined. We agree with the reviewer that 4-mm pitch on SEEG is not equivalent to 4-mm cortical recordings, however, we agree with the editor in thinking that the decoding performance from SEEG is a result that stands on its own and should be included and compared against μ ECoG.

We added following text to the introduction and discussion section:

Introduction:

Previous attempts to accurately decode speech have typically utilized invasive macro electrocorticography (macro ECoG, 10 mm inter-electrode spacing and 2.3 mm exposed diameter), and high-density ECoG (4 mm inter-electrode spacing and 1 mm exposed diameter), or stereo-electroencephalography (SEEG, 3.5 – 5 mm inter-electrode spacing), that target ventral sensorimotor cortex or speech motor cortex (SMC) during speech production^{2,8-11}.

Recordings of brain signals have previously been limited by ECoG recordings, which are typically measured from 64 – 128 contacts spaced 4 – 10 mm apart, or SEEG recordings that use depth probes (8 – 16 contacts) to measure cortical signals at 3.5 – 5 mm spatial resolution.

Discussion:

Finally, we directly compared the decoding ability of μ ECoG electrodes to standard IEEG (ECoG/SEEG) electrodes located with SMC. SEEG shanks contained depth electrodes (3.5 – 5 mm inter-electrode spacing) and the ECoG arrays contained electrodes (10 mm inter-electrode spacing) that were arranged in a rectangular lattice. In both cases, the electrodes that were directly present within/over SMC were utilized for decoding. We showed that even when with using short time periods, all three recording technologies obtained above-chance decoding performance, and micro-scale neural recordings using high-density μ ECoG outperformed current standard technology (Fig. 4c, and Supplementary Fig. 16). Our work is limited by not providing a direct comparison of decoding performance against high-density ECoG (4-mm inter-electrode

spacing), which has been previously been employed to achieve high-quality naturalistic speech synthesis⁶ and text generation⁷. Due to this limitation, we performed a subsampling analysis to simulate recordings at different spatial resolutions for a parametric and direction comparison. Future work will focus on using longer recordings, more naturalistic speech, and direct reconstruction of utterances from micro-scale neural recordings.

This anatomical clustering that we show is similar to previously identified somatotopic representations using high-density ECoG grids (4 mm inter-electrode spacing) that revealed a posterior clustering of lip articulators and an anterior clustering of tongue articulators with respect to central sulcus¹². Therefore, our high-density μ ECoG electrodes were able to maximally resolve these spatially confined articulatory 'hotspots', thereby, resulting in highly accurate decoding at less than 2 mm spatial resolution.

The authors have addressed all of my comments and clarified all my concerns. I have one minor suggestion regarding a change that was added in the revised version: Supplementary Figure 3 has 4 empty panels (I-L) indicating electrode-specific analysis regarding the absence of contamination in the HG band across subjects. I do not think that any of these panels add any additional value and suggest describing the absence purely in the text/caption.

We thank the reviewer for their valuable feedback. We removed the empty panels and added text to the legend describing the absence of acoustic contamination.

This paper assesses the contribution of μ ECoG recordings to improved decoding of speech from intracranial signals in sensorimotor cortex. The authors have adequately addressed my concerns in this revision, and have included important methodological details and clarifications, as well as several figures with information about localization and specific contrasts of interest. I have no further concerns.

We thank the reviewer for their valuable feedback.

1. Herff, C. & Schultz, T. Automatic speech recognition from neural signals: a focused review. *Front. Neurosci.* **10**, 429 (2016).
2. Herff, C., Krusienski, D. J. & Kubben, P. The potential of stereotactic-EEG for brain-computer interfaces: current progress and future directions. *Front. Neurosci.* **14**, 123 (2020).
3. Verwoert, M. *et al.* Dataset of Speech Production in intracranial Electroencephalography. *Sci. Data* **9**, 434 (2022).
4. Angrick, M. *et al.* Real-time synthesis of imagined speech processes from minimally invasive recordings of neural activity. *Commun. Biol.* **4**, 1055 (2021).
5. Pei, X., Barbour, D. L., Leuthardt, E. C. & Schalk, G. Decoding vowels and consonants in spoken and imagined words using electrocorticographic signals in humans. *J. Neural Eng.* **8**, 46028 (2011).
6. Anumanchipalli, G. K., Chartier, J. & Chang, E. F. Speech synthesis from neural decoding of spoken sentences. *Nature* **568**, 493–498 (2019).
7. Moses, D. A. *et al.* Neuroprosthesis for decoding speech in a paralyzed person with anarthria. *N. Engl. J. Med.* **385**, 217–227 (2021).
8. Mugler, E. M. *et al.* Differential Representation of Articulatory Gestures and Phonemes in Motor, Premotor, and Inferior Frontal Cortices. *J. Neurosci.* **4653**, 1–23 (2017).
9. Jiang, W., Pailla, T., Dichter, B., Chang, E. F. & Gilja, V. Decoding speech using the timing of neural

signal modulation. *Proc. Annu. Int. Conf. IEEE Eng. Med. Biol. Soc. EMBS 2016-October*, 1532–1535 (2016).

10. Cogan, G. B. *et al.* Manipulating stored phonological input during verbal working memory. *Nat. Neurosci.* **20**, 279–286 (2017).
11. Cogan, G. B. *et al.* Sensory-motor transformations for speech occur bilaterally. *Nature* **507**, 94–8 (2014).
12. Chartier, J., Anumanchipalli, G. K., Johnson, K. & Chang, E. F. Encoding of articulatory kinematic trajectories in human speech sensorimotor cortex. *Neuron* **98**, 1042–1054 (2018).